

# An analytical model investigating the impact of current shear and topographic fluctuations on surface waves

Meng Sun[1,2,3], Yongzeng Yang[1,2,3], Xunqiang Yin[1,2,3], Jisheng Ding[1], Tianqi Sun[4], Nan Jia[5]

[1]First Institute of Oceanography, Ministry of Natural Resources, Qingdao, 266061, China
[2]Laoshan Laboratory, Qingdao, 266237, China
[3]Key Laboratory of Marine Science and Numerical Modeling (MASNUM), Ministry of Natural Resources, Qingdao, 266061, China
[4]Institute of Remote Sensing, Navy Submarine Academy, Qingdao, 266000, China
[5]Institute of Geospatial Information, Information Engineering University, Zhengzhou, 450001, China

*Correspondence to*: Xunqiang Yin (yinxq@fio.org.cn)

**Abstract.** There are interactions between surface waves and currents, the latter includes wind-driven current and tidal current. Currents influence surface waves though advection transport and shear instability generation processes. While the horizontal gradient of current is commonly considered to calculate wave-current interaction source term in most wave models, the vertical gradient of current (current shear) has been simplified. In coastal waters, strong background currents with topographic fluctuations at the scale of surface waves have a resonance effect on surface waves. However, this resonance process is currently ignored in existing wave models. To evaluate the effects of current shear and topographic fluctuations on surface waves more accurately, an analytical model is proposed to describe the modification of the amplitude of orbital velocities for surface waves. The amplitude of orbital velocities exhibits significant variations when considering both current shear and topographic fluctuation effects. Wave particle trajectory equations that incorporate current shear and topographic fluctuations are derived based on this analytical model. In deep waters, current shear can increase or decrease the horizontal radius of wave particle trajectory by approximately 0.3m, while the modification of horizontal amplitude of orbital velocities is about 0.3m/s. In shallow waters, with both topographic fluctuations and background current present, both horizontal and vertical radii of wave particle trajectory change by approximately ±0.1m respectively, and the modification of both horizontal and vertical amplitudes of orbital velocities is about 0.2-0.3m/s. Moreover, in some cases, there are reversals in the direction of wave particle trajectories.

## 1 Introduction

Interactions between ocean motions, including turbulence, waves, eddies and circulation, occur at various spatial and temporal scales. The intricate nature of these interactions is crucial for understanding the dynamics of the ocean and accurately predicting its behavior. Among these processes, wave-current interactions play a significant role by facilitating



energy and momentum exchange between waves and currents. Furthermore, these interactions significantly influence sediment transport patterns and offshore structures design. A comprehensive understanding of wave-current interactions is imperative for advancing the development of more precise numerical models. China possesses extensive offshore waters The

combined effects of topography, tides, tidal currents, intense typhoons, and strong cold waves give rise to background flow fields characterized by a velocity of exceeding 100 cm/s and a spatial scale of nearly 100 km. In such circumstances, it becomes imperative to consider the intricate interactions between waves and currents.

The influence of surface waves on currents primarily manifests through wave breaking and the mixing processes of transport

flux residue induced by non-breaking surface waves and surface-wave-generated turbulence. The transport flux residue is a fundamental concept in fluid mechanics that describes the Reynolds average for advection transport. It is commonly used to quantify the transport volume of a specific physical quantity through a defined area per unit time during fluid flow. (1) The first mixing mechanism: The mixing of transport flux residue induced by non-breaking surface waves primarily encompasses the momentum mixing effect on ocean currents and the stirring mixing effect on temperature and salinity, which exerts a

significant influence in regions characterized by strong shear of currents and density (Yang et al., 2019; Shi et al., 2016; Yu et al., 2020; Zhuang et al., 2022, 2023). Radiation stress introduced by Longuet-Higgins and Stewar (1962, 1964), widely employed for wave-induced flow in shallow water, constitutes one of the terms contributing to the momentum mixing of surface waves with circulation. (2) The second mixing mechanism: Yuan et al. (1999, 2013) proposed an analytical estimation of the mixing coefficients induced by surface-wave-generated turbulence based on the equilibrium solutions of

the second-order turbulence closure model. This theory has been tested and confirmed through mechanically generate laboratory wave experiments (Babanin, 2006; Dai et al., 2010). The observed significant mixing strength plays an crucial role in the upper layer (Yang et al., 2003, 2004; Qiao et al., 2004, 2008, 2010; Xia et al., 2004, 2006). Yang et al. (2019) proposed analogue Reynolds Numbers for comparison between the first and the second mixing mechanisms. These two mixing mechanisms are comparable under typhoon conditions, and the first one performs remarkable in coastal areas. (3)

The third mixing mechanism: The breaking of waves can significantly impact the stability and intensity of the surface layer, resulting in more intense mixing and more uniform temperature distribution in this layer. Wave breaking not only generates downward energy flux but also introduces a nonzero length scale to the surface boundary conditions of the turbulence closure model (Agrawal et al., 1992; Craig and Banner, 1994; Terray et al., 1996, 1997, 1999; Sun et al., 2006). However, the mixing induced by breaking waves seems insufficient in ocean modeling (Burchard, 2001; Huang et al., 2011).


The influence of currents on surface waves primarily manifests through the processes of advection transport and shear instability generation. The current, serving as the background field for wave propagation, not only influences the group velocity and refractive direction of waves but also facilitates energy exchange with them. MASNUM wave model, incorporating characteristic inlaid scheme, represents a third generation wave model (Yuan et al., 1991a, 1991b; Yang et al.,



2005). The total mean wave energy balance equation used in MASNUM wave model, which is derived from the motion equations and boundary conditions of incompressible viscous fluid, is as follows:

$$\frac{\partial E}{\partial t} + \frac{\partial}{\partial x_\alpha}(U_\alpha E + F_\alpha) = -S_{\alpha\beta}\frac{\partial U_\beta}{\partial x_\alpha} + \epsilon_{sb} - \epsilon_{ds}$$

where $E$ is average total wave energy per unit cross-sectional area of the vertical water column, $F_\alpha$ is wave energy flux transferred by waves. $-S_{\alpha\beta}\frac{\partial U_\beta}{\partial x_\alpha}$ is the product of wave momentum flux tensor (i.g. radiation stress) and deformation tensor of ambient current, i.g. the wave-current interaction source function $S_{cu}$. $\epsilon_{sb}$ is the rate of the work done by external forces

on the surface and bottom. $\epsilon_{ds}$ is the energy loss rate due to internal viscosity. In the wave energy spectrum balance equation,

$$S_{cu} = -S_{\alpha\beta}\frac{\partial U_\beta}{\partial x_\alpha} = -\left\{\left[\frac{C_g}{C}(1+\cos^2\theta_1) - \frac{1}{2}\right]\frac{\partial U_x}{\partial x} + \frac{C_g}{C}\sin\theta_1\cos\theta_1\left(\frac{\partial U_x}{\partial y} + \frac{\partial U_y}{\partial x}\right) + \left[\frac{C_g}{C}(1+\sin^2\theta_1) - \frac{1}{2}\right]\frac{\partial U_y}{\partial y}\right\}E(\boldsymbol{k})$$

$$\alpha, \beta = 1, 2$$

where $E(\boldsymbol{k})$ is the wave-number spectrum. Evidently, the consideration of horizontal current gradient is accompanied by a simplification of vertical gradient (current shear). Ocean wave models, such as WAVEWATCH (Tolman, 1991), WAM4 (Günther, et al., 1992), SWAN (Booij, et al., 1999), are based on the wave action balance equation. Wave action density $N = E/\sigma$, introduced by Whitham (1965) and Brethrton and Garrett (1968), is conserved during propagation along its wave

characteristic in the presence of an ambient current. Therefore, the inclusion of a wave-current interaction source function is unnecessary in these three wave models. However, the consideration of vertical shears of current is also absent in these models. The analysis of the impact of current shear on surface waves constitutes one of primary focal points of this study.

The topography of offshore waters is intricate. The spatial-varied topography, characterized by diverse elevations and

fluctuations resembling a superposition of multiple terrains at various spatial scales. When strong currents generated by typhoons or tides flow through the seafloor near the shore, they encounter topographic fluctuations, resulting in the generation of water fluctuations at various spatial scales. There are resonance effects between water fluctuations exhibiting the same space-time scale as surface waves and the surface waves themselves. This resonance phenomenon induces alterations in the dynamics of pre-existing surface waves. This mechanism is disregarded in the majority of numerical wave

models. The evaluation and quantification of the impact of the mechanism on surface waves constitutes the other focal point of this study.

To address these two challenges, a fundamental theoretical framework is indispensable. A unified linear theory of wavelike perturbations (for surface gravity waves, internal gravity waves and inertial gravity waves) under general ocean conditions

(including large-scale current shear, density stratification, frontal zone, and arbitrary topography) is proposed by Yuan et al. (2011). The assumptions commonly accepted in the previous literature regarding the derivations of wave motions, such as the irrotationality assumption for surface gravity waves, the rigid-lid approximation for internal gravity waves, and the long-wave approximation for inertial gravity waves, are dismissed within the framework of this theory. The solutions of the



linearized instability analysis model, with five unknowns (three velocity components, pressure, density), were obtained

through Fourier integrals. The unified theory successfully reproduced classical results for both surface waves and internal waves under idealized conditions. The unified wave theory can serve as the theoretical basis for the dynamical explanation of surface waves in presence of large-scale current.

The derivation of the unified wave theory (Yuan et al., 2011) focuses on the Kuroshio region; therefore, only currents along

$x_2$-axis are considered, i.g. $\boldsymbol{U} = U(0, U_2, 0)$. The simplification of the gradient of topography is incorporated into the derivation. The present study addresses both of the aforementioned issues in a comprehensive manner. Furthermore, we employ Fourier integral of topography to determine the complete solutions. This study proposes an analytical model to quantitatively depict the amplitude variation of orbital velocities for surface waves in presence of background currents and topographic fluctuations. The effects of current shears and topographic fluctuations on wave orbital trajectories in both deep

and shallow waters are presented.

This paper is structured as follows. Section 2.1 presents the analytical model. Section 2.2 describes the multi-beam bathymetric data near Qingdao. In Section 3.1, we demonstrate the influence of current shears on surface waves in deep water. Furthermore, Section 3.2 presents an analysis of the impact of topographic fluctuations and current shears on surface

waves in shallow water. Finally, comprehensive conclusions are provided in Section 4.

## 2 Methodology and data

The amplitude variation of orbital velocities for surface waves is derived in this section through the development of an analytical model based motion equations and boundary conditions. This analytical model takes into account background currents and topographic fluctuations. A crucial parameter associated with topographic fluctuations in the analytical model is

determined using multi-beam terrain data in the vicinity of Qingdao.

### 2.1 Analytical Model Derivation

### 2.1.1 Solutions for gravity waves

The adiabatic governing equations, as formulated under the Boussinesq approximation (Yuan et al., 2011), can be expressed as follows.

The motion equations:

$$\frac{1}{R}u_1 + \frac{\partial u_1}{\partial x_1} + \frac{\partial u_2}{\partial x_2} + \frac{\partial u_3}{\partial x_3} = 0 \tag{1}$$





$$\frac{\partial u_1}{\partial t} + u_1 \frac{\partial u_1}{\partial x_1} + u_2 \frac{\partial u_1}{\partial x_2} + u_3 \frac{\partial u_1}{\partial x_3} - \frac{u_2^2}{R} - f u_2 = -\frac{\partial}{\partial x_1}\left(\frac{p}{\rho_0}\right) \tag{2}$$

$$\frac{\partial u_2}{\partial t} + u_1 \frac{\partial u_2}{\partial x_1} + u_2 \frac{\partial u_2}{\partial x_2} + u_3 \frac{\partial u_2}{\partial x_3} + \frac{u_1 u_2}{R} + f u_1 = -\frac{\partial}{\partial x_2}\left(\frac{p}{\rho_0}\right) \tag{3}$$

$$\frac{\partial u_3}{\partial t} + u_1 \frac{\partial u_3}{\partial x_1} + u_2 \frac{\partial u_3}{\partial x_2} + u_3 \frac{\partial u_3}{\partial x_3} = -\frac{\partial}{\partial x_3}\left(\frac{p}{\rho_0}\right) - g\left(\frac{\rho}{\rho_0}\right) \tag{4}$$

$$\frac{\partial}{\partial t}\left(\frac{\rho}{\rho_0}\right) + u_1 \frac{\partial}{\partial x_1}\left(\frac{\rho}{\rho_0}\right) + u_2 \frac{\partial}{\partial x_2}\left(\frac{\rho}{\rho_0}\right) + u_3 \frac{\partial}{\partial x_3}\left(\frac{\rho}{\rho_0}\right) = 0 \tag{5}$$

The boundary conditions:

$$\left(\frac{p}{\rho_0}\right)_{x_3=\zeta} = \left(\frac{p_A}{\rho_0}\right) \tag{6}$$

$$(u_3)_{x_3=\zeta} - \left[\frac{\partial \zeta}{\partial t} + \frac{\partial \zeta}{\partial x_1}(u_1)_{x_3=\zeta} + \frac{\partial \zeta}{\partial x_2}(u_2)_{x_3=\zeta}\right] = 0 \tag{7}$$

$$(u_3)_{x_3=-H} + \frac{\partial H}{\partial x_1}(u_1)_{x_3=-H} + \frac{\partial H}{\partial x_2}(u_2)_{x_3=-H} = 0 \tag{8}$$

In this coordinate system, the $x_1$ axis represents the eastward direction, while the $x_2$ axis denotes the northward direction. The symbols $\{u_1, u_2, u_3, p, \rho, \zeta\}$ represent the velocity components, pressure, density and surface elevation respectively. $f$ denotes the Coriolis parameter. $R$ is the curvature radius of the mean-flow path. $\rho_0$ denotes the basin mean water density and $p_A$ denotes the surface air pressure. $H$ represents the water depth.


Motion variables $\{u_1, u_2, u_3, p, \rho, \zeta\}$ can be categorized into mean motion $\{U_1, U_2, U_3, P, \bar{\rho}, Z\}$ and perturbations $\{u_{SM1}, u_{SM2}, u_{SM3}, p_{SM}, \rho_{SM}, h\}$. The subscript SM represents wave-like ocean motion. The function of water depth $H(x_1, x_2)$ is decomposed into two components: the slowly changing terrain at the scale of mean motion $\bar{H}(x_1, x_2)$ and the rapidly

changing terrain at wave scale $H_{SM}(x_1, x_2)$. Substituting for the mean flow variables $\{U_1, U_2, U_3, P, \bar{\rho}, Z\}$ into equation (1)-(8), we obtain the governing equations for the mean flow.

$$\frac{1}{R}U_1 + \frac{\partial U_1}{\partial x_1} + \frac{\partial U_2}{\partial x_2} + \frac{\partial U_3}{\partial x_3} = 0 \tag{9}$$

$$\frac{\partial U_1}{\partial t} + U_1 \frac{\partial U_1}{\partial x_1} + U_2 \frac{\partial U_1}{\partial x_2} + U_3 \frac{\partial U_1}{\partial x_3} - \bar{F}U_2 = -\frac{\partial}{\partial x_1}\left(\frac{P}{\rho_0}\right) \tag{10}$$





$$\frac{\partial U_2}{\partial t} + U_1 \frac{\partial U_2}{\partial x_1} + U_2 \frac{\partial U_2}{\partial x_2} + U_3 \frac{\partial U_2}{\partial x_3} + \overline{F} U_1 = -\frac{\partial}{\partial x_2}\left(\frac{P}{\rho_0}\right) \tag{11}$$

$$0 = -\frac{\partial}{\partial x_3}\left(\frac{P}{\rho_0}\right) - g\left(\frac{\overline{\rho}}{\rho_0}\right) \tag{12}$$

$$\frac{\partial}{\partial t}\left(\frac{\overline{\rho}}{\rho_0}\right) + U_1 \frac{\partial}{\partial x_1}\left(\frac{\overline{\rho}}{\rho_0}\right) + U_2 \frac{\partial}{\partial x_2}\left(\frac{\overline{\rho}}{\rho_0}\right) + U_3 \frac{\partial}{\partial x_3}\left(\frac{\overline{\rho}}{\rho_0}\right) = 0 \tag{13}$$

$$\left(\frac{P}{\rho_0}\right)_{x_3=0} = \left(\frac{P_A}{\rho_0}\right) + g\left(\frac{\overline{\rho}}{\rho_0}\right)_{x_3=0} Z \tag{14}$$

$$(U_3)_{x_3=0} - \left[\frac{\partial Z}{\partial t} + \frac{\partial Z}{\partial x_1}(U_1)_{x_3=0} + \frac{\partial Z}{\partial x_2}(U_2)_{x_3=0}\right] = 0 \tag{15}$$

$$(U_3)_{x_3=-H} + \frac{\partial \overline{H}}{\partial x_1}(U_1)_{x_3=-H} + \frac{\partial \overline{H}}{\partial x_2}(U_2)_{x_3=-H} = 0 \tag{16}$$

Where, $\overline{F} = f + \frac{U_2}{R}$.

Accordingly, by substituting the sum of the mean and perturbation variables into equations (1)-(8) and subtracting equations
(9)-(16) for the mean flow, we deduce the governing equations for the perturbations.

$$\frac{1}{R} u_{SM1} + \frac{\partial u_{SM1}}{\partial x_1} + \frac{\partial u_{SM2}}{\partial x_2} + \frac{\partial u_{SM3}}{\partial x_3} = 0 \tag{17}$$

$$\frac{\partial u_{SM1}}{\partial t} + U_1 \frac{\partial u_{SM1}}{\partial x_1} + U_2 \frac{\partial u_{SM1}}{\partial x_2} + U_3 \frac{\partial u_{SM1}}{\partial x_3} + u_{SM1} \frac{\partial U_1}{\partial x_1} + u_{SM2} \frac{\partial U_1}{\partial x_2} + u_{SM3} \frac{\partial U_1}{\partial x_3} - \overline{F} u_{SM2} - \frac{U_2}{R} u_{SM2} = -\frac{\partial}{\partial x_1}\left(\frac{p_{SM}}{\rho_0}\right) \tag{18}$$

$$\frac{\partial u_{SM2}}{\partial t} + U_1 \frac{\partial u_{SM2}}{\partial x_1} + U_2 \frac{\partial u_{SM2}}{\partial x_2} + U_3 \frac{\partial u_{SM2}}{\partial x_3} + u_{SM1} \frac{\partial U_2}{\partial x_1} + u_{SM2} \frac{\partial U_2}{\partial x_2} + u_{SM3} \frac{\partial U_2}{\partial x_3} + \overline{F} u_{SM1} + \frac{U_1}{R} u_{SM2} = -\frac{\partial}{\partial x_2}\left(\frac{p_{SM}}{\rho_0}\right) \tag{19}$$

$$\frac{\partial u_{SM3}}{\partial t} + U_1 \frac{\partial u_{SM3}}{\partial x_1} + U_2 \frac{\partial u_{SM3}}{\partial x_2} + U_3 \frac{\partial u_{SM3}}{\partial x_3} + u_{SM1} \frac{\partial U_3}{\partial x_1} + u_{SM2} \frac{\partial U_3}{\partial x_2} + u_{SM3} \frac{\partial U_3}{\partial x_3} = -\frac{\partial}{\partial x_3}\left(\frac{p_{SM}}{\rho_0}\right) - g\left(\frac{\rho_{SM}}{\rho_0}\right) \tag{20}$$

$$\frac{\partial}{\partial t}\left(\frac{\rho_{SM}}{\rho_0}\right) + U_1 \frac{\partial}{\partial x_1}\left(\frac{\rho_{SM}}{\rho_0}\right) + U_2 \frac{\partial}{\partial x_2}\left(\frac{\rho_{SM}}{\rho_0}\right) + U_3 \frac{\partial}{\partial x_3}\left(\frac{\rho_{SM}}{\rho_0}\right) + u_{SM1} \frac{\partial}{\partial x_1}\left(\frac{\overline{\rho}}{\rho_0}\right) + u_{SM2} \frac{\partial}{\partial x_2}\left(\frac{\overline{\rho}}{\rho_0}\right) + u_{SM3} \frac{\partial}{\partial x_3}\left(\frac{\overline{\rho}}{\rho_0}\right) = 0 \tag{21}$$

$$\left(\frac{p_{SM}}{\rho_0}\right)_{x_3=0} = \left(\frac{p'_A}{\rho_0}\right) + g\left(\frac{\overline{\rho}}{\rho_0}\right)_{x_3=0} h \tag{22}$$





$$(u_{SM3})_{x_3=0} - \left[\frac{\partial h}{\partial t} + \frac{\partial Z}{\partial x_1}(u_{SM1})_{x_3=0} + \frac{\partial Z}{\partial x_2}(u_{SM2})_{x_3=0} + \frac{\partial h}{\partial x_1}(U_1)_{x_3=0} + \frac{\partial h}{\partial x_2}(U_2)_{x_3=0}\right] = 0 \tag{23}$$

$$(u_{SM3})_{x_3=-H} + \frac{\partial H}{\partial x_1}(u_{SM1})_{x_3=-H} + \frac{\partial H}{\partial x_2}(u_{SM2})_{x_3=-H} + \frac{\partial H_{SM}}{\partial x_1}(U_1)_{x_3=-H} + \frac{\partial H_{SM}}{\partial x_2}(U_2)_{x_3=-H} = 0 \tag{24}$$

In equations (18-21), the blue terms represent the transport of background current, while the green terms denote the shear generation of background current. The transport terms are eliminated through coordinate transformation $\{x'_1 = x_1 - U_1 t, x'_2 = x_2 - U_2 t, x'_3 = x_3 - U_3 t, t' = t\}$ for the purpose of simplification. Thus we obtain the governing equations for

the perturbations (with apostrophes omitted).

$$\frac{1}{R}u_{SM1} + \frac{\partial u_{SM1}}{\partial x_1} + \frac{\partial u_{SM2}}{\partial x_2} + \frac{\partial u_{SM3}}{\partial x_3} = 0 \tag{25}$$

$$\frac{\partial u_{SM1}}{\partial t} + u_{SM1}\frac{\partial U_1}{\partial x_1} + u_{SM2}\frac{\partial U_1}{\partial x_2} + u_{SM3}\frac{\partial U_1}{\partial x_3} - \overline{F}u_{SM2} - \frac{U_2}{R}u_{SM2} = -\frac{\partial}{\partial x_1}\left(\frac{p_{SM}}{\rho_0}\right) \tag{26}$$

$$\frac{\partial u_{SM2}}{\partial t} + u_{SM1}\frac{\partial U_2}{\partial x_1} + u_{SM2}\frac{\partial U_2}{\partial x_2} + u_{SM3}\frac{\partial U_2}{\partial x_3} + \overline{F}u_{SM1} + \frac{U_1}{R}u_{SM2} = -\frac{\partial}{\partial x_2}\left(\frac{p_{SM}}{\rho_0}\right) \tag{27}$$

$$\frac{\partial u_{SM3}}{\partial t} + u_{SM1}\frac{\partial U_3}{\partial x_1} + u_{SM2}\frac{\partial U_3}{\partial x_2} + u_{SM3}\frac{\partial U_3}{\partial x_3} = -\frac{\partial}{\partial x_3}\left(\frac{p_{SM}}{\rho_0}\right) - g\left(\frac{p_{SM}}{\rho_0}\right) \tag{28}$$

$$\frac{\partial}{\partial t}\left(\frac{\rho_{SM}}{\rho_0}\right) + u_{SM1}\frac{\partial}{\partial x_1}\left(\frac{\overline{\rho}}{\rho_0}\right) + u_{SM2}\frac{\partial}{\partial x_2}\left(\frac{\overline{\rho}}{\rho_0}\right) + u_{SM3}\frac{\partial}{\partial x_3}\left(\frac{\overline{\rho}}{\rho_0}\right) = 0 \tag{29}$$

$$\left(\frac{p_{SM}}{\rho_0}\right)_{x_3=0} = \left(\frac{p'_A}{\rho_0}\right) + g\left(\frac{\overline{\rho}}{\rho_0}\right)_{x_3=0} h \tag{30}$$

$$(u_{SM3})_{x_3=0} - \left[\frac{\partial h}{\partial t} + \frac{\partial Z}{\partial x_1}(u_{SM1})_{x_3=0} + \frac{\partial Z}{\partial x_2}(u_{SM2})_{x_3=0}\right] = 0 \tag{31}$$

$$(u_{SM3})_{x_3=-H} + \frac{\partial H}{\partial x_1}(u_{SM1})_{x_3=-H} + \frac{\partial H}{\partial x_2}(u_{SM2})_{x_3=-H} + \frac{\partial H_{SM}}{\partial x_1}(U_1)_{x_3=-H} + \frac{\partial H_{SM}}{\partial x_2}(U_2)_{x_3=-H} = 0 \tag{32}$$

In the sense of the generalized function, any arbitrary function can be written as the Fourier integral forms. Therefore, we express the perturbations variables as their Fourier integrals in the following forms:

$$u_{SM1} = \iint_{k_1,k_2} \mu_1(x_3)\exp\{i(k_1 x_1 + k_2 x_2 - \omega t)\}dk_1 dk_2 \qquad \left(\frac{p_{SM}}{\rho_0}\right) = \iint_{k_1,k_2} \phi(x_3)\exp\{i(k_1 x_1 + k_2 x_2 - \omega t)\}dk_1 dk_2 \tag{33}$$





$$u_{SM2} = \iint\limits_{k_1,k_2} \mu_2(x_3)\exp\{i(k_1x_1 + k_2x_2 - \omega t)\}dk_1 dk_2 \qquad \left(\frac{\rho_{SM}}{\rho_0}\right) = \iint\limits_{k_1,k_2} \beta(x_3)\exp\{i(k_1x_1 + k_2x_2 - \omega t)\}dk_1 dk_2$$

$$u_{SM3} = \iint\limits_{k_1,k_2} \mu_3(x_3)\exp\{i(k_1x_1 + k_2x_2 - \omega t)\}dk_1 dk_2 \qquad h = \iint\limits_{k_1,k_2} \eta\exp\{i(k_1x_1 + k_2x_2 - \omega t)\}dk_1 dk_2$$

$$H_{SM} = \iint\limits_{k_1,k_2} \mu_{(H_{SM})}\exp\{i(k_1x_1 + k_2x_2 - \omega t)\}dk_1 dk_2 \tag{34}$$

The real part $\omega_R$ of the complex frequency $\omega = \omega_R + i\omega_I$ is interpreted as the physical frequency in this context. Given the much larger spatial scale of the mean flow compared to the perturbation flow, the parameters related to the mean flow are considered invariant when then Fourier integrals (33)-(34) are integrated into the governing equations (25)-(32). In consequence, we derive the governing equations for the perturbations flow in the phase space, as follows. Here $\mu_{(H_{SM})}$ represents the coefficient in Fourier inverse transformation of fast changing terrain function $H_{SM}$.

$$(\gamma + ik_1)\mu_1 + ik_2\mu_2 + \frac{\partial \mu_3}{\partial x_3} = 0 \tag{35}$$

$$-i\omega\mu_1 + \mu_1\frac{\partial U_1}{\partial x_1} + \mu_2\frac{\partial U_1}{\partial x_2} + \mu_3\frac{\partial U_1}{\partial x_3} - \overline{F}\mu_2 - \frac{U_2}{R}\mu_2 = -ik_1\phi \tag{36}$$

$$-i\omega\mu_2 + \mu_1\frac{\partial U_2}{\partial x_1} + \mu_2\frac{\partial U_2}{\partial x_2} + \mu_3\frac{\partial U_2}{\partial x_3} + \overline{F}\mu_1 + \frac{U_1}{R}\mu_2 = -ik_2\phi \tag{37}$$

$$-i\omega\mu_3 + \mu_1\frac{\partial U_3}{\partial x_1} + \mu_2\frac{\partial U_3}{\partial x_2} + \mu_3\frac{\partial U_3}{\partial x_3} = -\frac{\partial \phi}{\partial x_3} - g\beta \tag{38}$$

$$\beta = i\frac{1}{g\omega}\left(M_1^2\mu_1 + M_2^2\mu_2 + N^2\mu_3\right) \tag{39}$$

$$(\phi)_{x_3=0} = \phi_A + g\left(\frac{\overline{p}}{\rho_0}\right)_{x_3=0}\eta \tag{40}$$

$$(\mu_3)_{x_3=0} = -i\omega\eta + \frac{\partial Z}{\partial x_1}(\mu_1)_{x_3=0} + \frac{\partial Z}{\partial x_2}(\mu_2)_{x_3=0} \tag{41}$$

$$(\mu_3)_{x_3=-H} + \frac{\partial H}{\partial x_1}(\mu_1)_{x_3=-H} + \frac{\partial H}{\partial x_2}(\mu_2)_{x_3=-H} + ik_1\mu_{(H_{SM})}(U_1)_{x_3=-H} + ik_2\mu_{(H_{SM})}(U_2)_{x_3=-H} = 0 \tag{42}$$

$$\gamma = \frac{1}{R} \qquad M_1^2 = -g\frac{\partial}{\partial x_1}\left(\frac{\overline{p}}{\rho_0}\right) \qquad M_2^2 = -g\frac{\partial}{\partial x_2}\left(\frac{\overline{p}}{\rho_0}\right) \qquad N^2 = -g\frac{\partial}{\partial x_3}\left(\frac{\overline{p}}{\rho_0}\right) \tag{43}$$






Next, we will proceed to find the solutions for the equations pertaining to the Fourier transformation functions.

**Step 1.1**: By deriving from the algebraic equations (36) and (37), we obtain equations (45) and (46).

**Step 1.2**: After substitution of equations (45) and (46) into equations (35), (38) and (39), and subsequent manipulation, we obtain equations (44), (47) and (48).

**Step 1.3**: After substituting equations (45) and (46) into equation (42), we obtain equation (51).

$$-i\omega k\left\{\frac{\partial U_1}{\partial x_3}\Pi_{31} + \frac{\partial U_2}{\partial x_3}\Pi_{32}\right\}\mu_3 + \omega k^2\Pi_4\phi - i\Omega^2\frac{\partial \mu_3}{\partial x_3} = 0 \tag{44}$$

$$\mu_1 = \frac{1}{\Omega^2}\left[\frac{\partial U_2}{\partial x_3}\left(\overline{F} + \frac{U_2}{R} - \frac{\partial U_1}{\partial x_2}\right) - i\frac{\partial U_1}{\partial x_3}\left(\omega + i\frac{U_1}{R} + i\frac{\partial U_2}{\partial x_2}\right)\right]\mu_3 + \frac{\omega k}{\Omega^2}\Pi_1\phi \tag{45}$$

$$\mu_2 = -\frac{1}{\Omega^2}\left[\frac{\partial U_1}{\partial x_3}\left(\overline{F} + \frac{\partial U_2}{\partial x_1}\right) + i\frac{\partial U_2}{\partial x_3}\left(\omega + i\frac{\partial U_1}{\partial x_1}\right)\right]\mu_3 + \frac{\omega k}{\Omega^2}\Pi_2\phi \tag{46}$$

$$\left\{\begin{array}{c}\omega^2 - \left(N^2\right)' + i\omega\frac{\partial U_3}{\partial x_3} \\ +\frac{\partial U_1}{\partial x_3}\frac{\partial U_3}{\partial x_1}\frac{\omega\left(\omega + i\frac{U_1}{R} + i\frac{\partial U_2}{\partial x_2}\right)}{\Omega^2} + i\frac{\partial U_2}{\partial x_3}\frac{\partial U_3}{\partial x_1}\frac{\omega\left(\overline{F} + \frac{U_2}{R} - \frac{\partial U_1}{\partial x_2}\right)}{\Omega^2} \\ +\frac{\partial U_3}{\partial x_2}\frac{\partial U_2}{\partial x_3}\frac{\omega\left(\omega + i\frac{\partial U_1}{\partial x_1}\right)}{\Omega^2} - i\frac{\partial U_1}{\partial x_3}\frac{\partial U_3}{\partial x_2}\frac{\omega\left(\overline{F} + \frac{\partial U_2}{\partial x_1}\right)}{\Omega^2}\end{array}\right\}\mu_3 + \frac{\omega k}{\Omega^2}\left\{\begin{array}{c}\left(-M_1^2 + i\omega\frac{\partial U_3}{\partial x_1}\right)\Pi_1 \\ +\left(-M_2^2 + i\omega\frac{\partial U_3}{\partial x_2}\right)\Pi_2\end{array}\right\}\phi + i\omega\frac{\partial \phi}{\partial x_3} = 0 \tag{47}$$

$$\beta = i\frac{1}{g\omega}\left\{\left(N^2\right)'\mu_3 + \frac{\omega k}{\Omega^2}\left(M_1^2\Pi_1 + M_2^2\Pi_2\right)\phi\right\} \tag{48}$$

$$(\phi)_{x_3=0} = \phi_A + g\left(\frac{\overline{P}}{\rho_0}\right)_{x_3=0}\eta \tag{49}$$

$$(\mu_3)_{x_3=0} = -i\omega\eta + \frac{\partial Z}{\partial x_1}(\mu_1)_{x_3=0} + \frac{\partial Z}{\partial x_2}(\mu_2)_{x_3=0} \tag{50}$$

$$\left\{\begin{array}{c}1 + \frac{1}{\Omega^2}\left[\frac{\partial U_2}{\partial x_3}\left(\overline{F} + \frac{U_2}{R} - \frac{\partial U_1}{\partial x_2}\right) - i\frac{\partial U_1}{\partial x_3}\left(\omega + i\frac{U_1}{R} + i\frac{\partial U_2}{\partial x_2}\right)\right]\frac{\partial H}{\partial x_1} \\ -\frac{1}{\Omega^2}\left[\frac{\partial U_1}{\partial x_3}\left(\overline{F} + \frac{\partial U_2}{\partial x_1}\right) + i\frac{\partial U_2}{\partial x_3}\left(\omega + i\frac{\partial U_1}{\partial x_1}\right)\right]\frac{\partial H}{\partial x_2}\end{array}\right\}_{x_3=-H}(\mu_3)_{x_3=-H}$$
$$+\frac{\omega k}{\Omega^2}\left\{\frac{\partial H}{\partial x_1}\Pi_1 + \frac{\partial H}{\partial x_2}\Pi_2\right\}_{x_3=-H}(\phi)_{x_3=-H} + ik_1\mu_{(H_{SM})}(U_1)_{x_3=-H} + ik_2\mu_{(H_{SM})}(U_2)_{x_3=-H} = 0 \tag{51}$$

$$k = \sqrt{k_1^2 + k_2^2} \tag{52}$$



$$\Omega^2 = \left(\omega + i\frac{\partial U_1}{\partial x_1}\right)\left(\omega + i\frac{U_1}{R} + i\frac{\partial U_2}{\partial x_2}\right) - \left(\overline{F} + \frac{\partial U_2}{\partial x_1}\right)\left(\overline{F} + \frac{U_2}{R} - \frac{\partial U_1}{\partial x_2}\right)$$

$$\Pi_1 = \frac{k_1}{k}\frac{\omega + i\frac{U_1}{R} + i\frac{\partial U_2}{\partial x_2}}{\omega} + i\frac{k_2}{k}\frac{\overline{F} + \frac{U_2}{R} - \frac{\partial U_1}{\partial x_2}}{\omega} \qquad \Pi_2 = -i\frac{k_1}{k}\frac{\overline{F} + \frac{\partial U_2}{\partial x_1}}{\omega} + \frac{k_2}{k}\frac{\omega + i\frac{\partial U_1}{\partial x_1}}{\omega}$$

$$\Pi_{31} = \frac{k_1 - i\gamma}{k}\frac{\omega + i\frac{U_1}{R} + i\frac{\partial U_2}{\partial x_2}}{\omega} - i\frac{k_2}{k}\frac{\overline{F} + \frac{\partial U_2}{\partial x_1}}{\omega} \qquad \Pi_{32} = \frac{k_2}{k}\frac{\omega + i\frac{\partial U_1}{\partial x_1}}{\omega} + i\frac{k_1 - i\gamma}{k}\frac{\overline{F} + \frac{U_2}{R} - \frac{\partial U_1}{\partial x_2}}{\omega}$$

$$\Pi_4 = \frac{k_1(k_1 - i\gamma)}{k^2}\frac{\omega + i\frac{U_1}{R} + i\frac{\partial U_2}{\partial x_2}}{\omega} + i\frac{(k_1 - i\gamma)k_2}{k^2}\frac{\overline{F} + \frac{U_2}{R} - \frac{\partial U_1}{\partial x_2}}{\omega} - i\frac{k_1 k_2}{k^2}\frac{\overline{F} + \frac{\partial U_2}{\partial x_1}}{\omega} + \frac{k_2^2}{k^2}\frac{\omega + i\frac{\partial U_1}{\partial x_1}}{\omega}$$

$$(N^2)' = N^2 + \frac{M_1^2}{\Omega^2}\left[\frac{\partial U_2}{\partial x_3}\left(\overline{F} + \frac{U_2}{R} - \frac{\partial U_1}{\partial x_2}\right) - i\frac{\partial U_1}{\partial x_3}\left(\omega + i\frac{U_1}{R} + i\frac{\partial U_2}{\partial x_2}\right)\right] - \frac{M_2^2}{\Omega^2}\left[\frac{\partial U_1}{\partial x_3}\left(\overline{F} + \frac{\partial U_2}{\partial x_1}\right) + i\frac{\partial U_2}{\partial x_3}\left(\omega + i\frac{\partial U_1}{\partial x_1}\right)\right]$$

**Step 2**: In order to establish a homogenous boundary condition, we employ the following transformation (53). After substitution of the aforementioned transformation into the motion equations (44)-(48) and the boundary conditions (49)- (51), we derived the governing equations (55)-(62) related to $\left(\overline{\mu}_3, \phi\right)$.

$$\overline{\mu}_3 = \mu_3 + \delta_{-H}\phi \tag{53}$$

$$\delta_{-H} = \frac{\omega k}{\Omega^2}\left\{\frac{\partial H}{\partial x_1}\Pi_1 + \frac{\partial H}{\partial x_2}\Pi_2\right\}_{x_3=-H} \left\{1 + \frac{1}{\Omega^2}\left[\frac{\partial U_2}{\partial x_3}\left(\overline{F} + \frac{U_2}{R} - \frac{\partial U_1}{\partial x_2}\right) - i\frac{\partial U_1}{\partial x_3}\left(\omega + i\frac{U_1}{R} + i\frac{\partial U_2}{\partial x_2}\right)\right]\frac{\partial H}{\partial x_1} - \frac{1}{\Omega^2}\left[\frac{\partial U_1}{\partial x_3}\left(\overline{F} + \frac{\partial U_2}{\partial x_1}\right) + i\frac{\partial U_2}{\partial x_3}\left(\omega + i\frac{\partial U_1}{\partial x_1}\right)\right]\frac{\partial H}{\partial x_2}\right\}_{x_3=-H}^{-1} \tag{54}$$


$$\left\{\begin{array}{l} -i\omega\left(\frac{\partial U_1}{\partial x_3}\Pi_{31} + \frac{\partial U_2}{\partial x_3}\Pi_{32}\right) \\ -\left\{\begin{array}{l}\Omega^2\left[\omega^2 - (N^2)' + i\omega\frac{\partial U_3}{\partial x_3}\right] \\ +\frac{\partial U_1}{\partial x_3}\frac{\partial U_3}{\partial x_1}\omega\left(\omega + i\frac{U_1}{R} + i\frac{\partial U_2}{\partial x_2}\right) + i\frac{\partial U_2}{\partial x_3}\frac{\partial U_3}{\partial x_1}\omega\left(\overline{F} + \frac{U_2}{R} - \frac{\partial U_1}{\partial x_2}\right) \\ +\frac{\partial U_3}{\partial x_2}\frac{\partial U_2}{\partial x_3}\omega\left(\omega + i\frac{\partial U_1}{\partial x_1}\right) - i\frac{\partial U_1}{\partial x_3}\frac{\partial U_3}{\partial x_2}\omega\left(\overline{F} + \frac{\partial U_2}{\partial x_1}\right)\end{array}\right\}\left(\frac{\delta_{-H}}{\omega k}\right)\end{array}\right\}\overline{\mu}_3 + \omega k(\Pi_4)'\phi - i\frac{\Omega^2}{k}\frac{\partial\overline{\mu}_3}{\partial x_3} = 0 \tag{55}$$

$$\mu_1 = \frac{1}{\Omega^2}\left[\begin{array}{l}\frac{\partial U_2}{\partial x_3}\left(\overline{F} + \frac{U_2}{R} - \frac{\partial U_1}{\partial x_2}\right) \\ -i\frac{\partial U_1}{\partial x_3}\left(\omega + i\frac{U_1}{R} + i\frac{\partial U_2}{\partial x_2}\right)\end{array}\right]\overline{\mu}_3 + \frac{\omega k}{\Omega^2}\left\{\Pi_1 - \left[\begin{array}{l}\frac{\partial U_2}{\partial x_3}\left(\overline{F} + \frac{U_2}{R} - \frac{\partial U_1}{\partial x_2}\right) \\ -i\frac{\partial U_1}{\partial x_3}\left(\omega + i\frac{U_1}{R} + i\frac{\partial U_2}{\partial x_2}\right)\end{array}\right]\left(\frac{\delta_{-H}}{\omega k}\right)\right\}\phi \tag{56}$$



$$\mu_2 = -\frac{1}{\Omega^2}\left[\begin{array}{c}\frac{\partial U_1}{\partial x_3}\left(\overline{F}+\frac{\partial U_2}{\partial x_1}\right)\\+i\frac{\partial U_2}{\partial x_3}\left(\omega+i\frac{\partial U_1}{\partial x_1}\right)\end{array}\right]\overline{\mu}_3+\frac{\omega k}{\Omega^2}\left\{\Pi_2+\left[\begin{array}{c}\frac{\partial U_1}{\partial x_3}\left(\overline{F}+\frac{\partial U_2}{\partial x_1}\right)\\+i\frac{\partial U_2}{\partial x_3}\left(\omega+i\frac{\partial U_1}{\partial x_1}\right)\end{array}\right]\left(\frac{\delta_{-H}}{\omega k}\right)\right\}\phi \tag{57}$$

$$\left\{\begin{array}{c}\Omega^2\left[\omega^2-(N^2)'+i\omega\frac{\partial U_3}{\partial x_3}\right]\\+\frac{\partial U_1}{\partial x_3}\frac{\partial U_3}{\partial x_1}\omega\left(\omega+i\frac{U_1}{R}+i\frac{\partial U_2}{\partial x_2}\right)+i\frac{\partial U_2}{\partial x_3}\frac{\partial U_3}{\partial x_1}\omega\left(\overline{F}+\frac{U_2}{R}-\frac{\partial U_1}{\partial x_2}\right)\\+\frac{\partial U_3}{\partial x_2}\frac{\partial U_2}{\partial x_3}\omega\left(\omega+i\frac{\partial U_1}{\partial x_1}\right)-i\frac{\partial U_1}{\partial x_3}\frac{\partial U_3}{\partial x_2}\omega\left(\overline{F}+\frac{\partial U_2}{\partial x_1}\right)\end{array}\right\}\overline{\mu}_3$$

$$-\omega k\left\{\begin{array}{c}\left(M_1^2-i\omega\frac{\partial U_3}{\partial x_1}\right)\Pi_1+\left(M_2^2-i\omega\frac{\partial U_3}{\partial x_2}\right)\Pi_2\\+\left\{\begin{array}{c}\Omega^2\left[\omega^2-(N^2)'+i\omega\frac{\partial U_3}{\partial x_3}\right]\\+\frac{\partial U_1}{\partial x_3}\frac{\partial U_3}{\partial x_1}\omega\left(\omega+i\frac{U_1}{R}+i\frac{\partial U_2}{\partial x_2}\right)+i\frac{\partial U_2}{\partial x_3}\frac{\partial U_3}{\partial x_1}\omega\left(\overline{F}+\frac{U_2}{R}-\frac{\partial U_1}{\partial x_2}\right)\\+\frac{\partial U_3}{\partial x_2}\frac{\partial U_2}{\partial x_3}\omega\left(\omega+i\frac{\partial U_1}{\partial x_1}\right)-i\frac{\partial U_1}{\partial x_3}\frac{\partial U_3}{\partial x_2}\omega\left(\overline{F}+\frac{\partial U_2}{\partial x_1}\right)\end{array}\right\}\left(\frac{\delta_{-H}}{\omega k}\right)\end{array}\right\}\phi+i\omega\Omega^2\frac{\partial\phi}{\partial x_3}=0 \tag{58}$$

$$\beta = i\frac{1}{g\omega}\left\{(N^2)'\overline{\mu}_3+\frac{\omega k}{\Omega^2}\left[M_1^2\Pi_1+M_2^2\Pi_2-\Omega^2(N^2)'\left(\frac{\delta_{-H}}{\omega k}\right)\right]\phi\right\} \tag{59}$$

$$(\phi)_{x_3=0} = \phi_A+g\left(\frac{\overline{p}}{\rho_0}\right)_{x_3=0}\eta \tag{60}$$

$$(\overline{\mu}_3)_{x_3=0} = -i\overline{\omega}\eta+\frac{\partial Z}{\partial x_1}(\mu_1)_{x_3=0}+\frac{\partial Z}{\partial x_2}(\mu_2)_{x_3=0}+\delta_{-H}\phi_A \tag{61}$$

$$(\overline{\mu}_3)_{x_3=-H} = \hat{\delta}_{-H} \tag{62}$$

$$(\Pi_4)' = \Pi_4+\left\{\begin{array}{c}i\omega\left(\frac{\partial U_1}{\partial x_3}\Pi_{31}+\frac{\partial U_2}{\partial x_3}\Pi_{32}\right)\\+\left(M_1^2-i\omega\frac{\partial U_3}{\partial x_1}\right)\Pi_1+\left(M_2^2-i\omega\frac{\partial U_3}{\partial x_2}\right)\Pi_2\end{array}\right\}\left(\frac{\delta_{-H}}{\omega k}\right)$$

$$+\left\{\begin{array}{c}\Omega^2\left[\omega^2-(N^2)'+i\omega\frac{\partial U_3}{\partial x_3}\right]\\+\frac{\partial U_1}{\partial x_3}\frac{\partial U_3}{\partial x_1}\omega\left(\omega+i\frac{U_1}{R}+i\frac{\partial U_2}{\partial x_2}\right)+i\frac{\partial U_2}{\partial x_3}\frac{\partial U_3}{\partial x_1}\omega\left(\overline{F}+\frac{U_2}{R}-\frac{\partial U_1}{\partial x_2}\right)\\+\frac{\partial U_3}{\partial x_2}\frac{\partial U_2}{\partial x_3}\omega\left(\omega+i\frac{\partial U_1}{\partial x_1}\right)-i\frac{\partial U_1}{\partial x_3}\frac{\partial U_3}{\partial x_2}\omega\left(\overline{F}+\frac{\partial U_2}{\partial x_1}\right)\end{array}\right\}\left(\frac{\delta_{-H}}{\omega k}\right)^2 \tag{63}$$

$$\overline{\omega} = \omega\left[1+igk\left(\frac{\overline{p}}{\rho_0}\right)_{x_3=0}\left(\frac{\delta_{-H}}{\omega_0 k}\right)\right]$$



$$\hat{\delta}_{-H} = -\begin{bmatrix} ik_1\mu_{(H_{SM})}(U_1)_{x_3=-H} \\ +ik_2\mu_{(H_{SM})}(U_2)_{x_3=-H} \end{bmatrix} \left\{ \begin{matrix} 1 + \frac{1}{\Omega^2}\left[\frac{\partial U_2}{\partial x_3}\left(\overline{F} + \frac{U_2}{R} - \frac{\partial U_1}{\partial x_2}\right) - i\frac{\partial U_1}{\partial x_3}\left(\omega + i\frac{U_1}{R} + i\frac{\partial U_2}{\partial x_2}\right)\right]\frac{\partial H}{\partial x_1} \\ - \frac{1}{\Omega^2}\left[\frac{\partial U_1}{\partial x_3}\left(\overline{F} + \frac{\partial U_2}{\partial x_1}\right) + i\frac{\partial U_2}{\partial x_3}\left(\omega + i\frac{\partial U_1}{\partial x_1}\right)\right]\frac{\partial H}{\partial x_2} \end{matrix} \right\}^{-1}_{x_3=-H}$$

**Step 3.1**: The Fourier transformation function $\overline{\mu}_3$ and $\phi$ can be expressed in a fluctuation form, as illustrated in formula (64). Here $\varepsilon_3$ is the slowly-changing coefficient in the vertical direction. Substitute equation (64) into equations (55) and (58), we obtain equations (65) and (66), which is related to $A$ and $B$.

$$\overline{\mu}_3 = A(\varepsilon_3 x_3)\exp\{iX_3(x_3)\} \qquad\qquad \phi = B(\varepsilon_3 x_3)\exp\{iX_3(x_3)\} \tag{64}$$

$$\left\{ \begin{matrix} -i\omega k\left(\frac{\partial U_1}{\partial x_3}\Pi_{31} + \frac{\partial U_2}{\partial x_3}\Pi_{32}\right) + \Omega^2\frac{\partial X_3}{\partial x_3} \\ -k\left\{ \begin{matrix} \Omega^2\left[\omega^2 - (N^2)' + i\omega\frac{\partial U_3}{\partial x_3}\right] \\ +\frac{\partial U_1}{\partial x_3}\frac{\partial U_3}{\partial x_1}\omega\left(\omega + i\frac{U_1}{R} + i\frac{\partial U_2}{\partial x_2}\right) + i\frac{\partial U_2}{\partial x_3}\frac{\partial U_3}{\partial x_1}\omega\left(\overline{F} + \frac{U_2}{R} - \frac{\partial U_1}{\partial x_2}\right) \\ +\frac{\partial U_3}{\partial x_2}\frac{\partial U_2}{\partial x_3}\omega\left(\omega + i\frac{\partial U_1}{\partial x_1}\right) - i\frac{\partial U_1}{\partial x_3}\frac{\partial U_3}{\partial x_2}\omega\left(\overline{F} + \frac{\partial U_2}{\partial x_1}\right) \end{matrix} \right\}\left(\frac{\delta_{-H}}{\omega k}\right) \end{matrix} \right\} A + \omega k^2(\Pi_4)'\boldsymbol{B} = 0 \tag{65}$$

$$\left\{ \begin{matrix} \Omega^2\left[\omega^2 - (N^2)' + i\omega\frac{\partial U_3}{\partial x_3}\right] \\ +\frac{\partial U_1}{\partial x_3}\frac{\partial U_3}{\partial x_1}\omega\left(\omega + i\frac{U_1}{R} + i\frac{\partial U_2}{\partial x_2}\right) + i\frac{\partial U_2}{\partial x_3}\frac{\partial U_3}{\partial x_1}\omega\left(\overline{F} + \frac{U_2}{R} - \frac{\partial U_1}{\partial x_2}\right) \\ +\frac{\partial U_3}{\partial x_2}\frac{\partial U_2}{\partial x_3}\omega\left(\omega + i\frac{\partial U_1}{\partial x_1}\right) - i\frac{\partial U_1}{\partial x_3}\frac{\partial U_3}{\partial x_2}\omega\left(\overline{F} + \frac{\partial U_2}{\partial x_1}\right) \end{matrix} \right\} A \\ -\omega\left\{ \begin{matrix} k\left[\left(M_1^2 - i\omega\frac{\partial U_3}{\partial x_1}\right)\Pi_1 + \left(M_2^2 - i\omega\frac{\partial U_3}{\partial x_2}\right)\Pi_2\right] + \Omega^2\frac{\partial X_3}{\partial x_3} \\ +k\left\{ \begin{matrix} \Omega^2\left[\omega^2 - (N^2)' + i\omega\frac{\partial U_3}{\partial x_3}\right] \\ +\frac{\partial U_1}{\partial x_3}\frac{\partial U_3}{\partial x_1}\omega\left(\omega + i\frac{U_1}{R} + i\frac{\partial U_2}{\partial x_2}\right) + i\frac{\partial U_2}{\partial x_3}\frac{\partial U_3}{\partial x_1}\omega\left(\overline{F} + \frac{U_2}{R} - \frac{\partial U_1}{\partial x_2}\right) \\ +\frac{\partial U_3}{\partial x_2}\frac{\partial U_2}{\partial x_3}\omega\left(\omega + i\frac{\partial U_1}{\partial x_1}\right) - i\frac{\partial U_1}{\partial x_3}\frac{\partial U_3}{\partial x_2}\omega\left(\overline{F} + \frac{\partial U_2}{\partial x_1}\right) \end{matrix} \right\}\left(\frac{\delta_{-H}}{\omega k}\right) \end{matrix} \right\}\boldsymbol{B} = 0 \tag{66}$$


**Step 3.2**: If the fluctuation discussed in this paper has a non-zero solution, then the determinant of the coefficient for amplitude parameters $A$ and $B$ is zero as shown in equation (67). Here the terms of $\delta_{-H}$ is omitted, which implies that the vertical wave number and the phase function of the perturbations are independent of the bottom topography which is reasonable.



$$a\left(\Omega^2 \frac{\partial X_3}{\partial x_3}\right)^2 + b\left(\Omega^2 \frac{\partial X_3}{\partial x_3}\right) + c = 0 \tag{67}$$

$$a = 1 \qquad b = k\left\{-i\omega\left(\frac{\partial U_1}{\partial x_3}\Pi_{31} + \frac{\partial U_2}{\partial x_3}\Pi_{32}\right) + \left(M_1^2 - i\omega\frac{\partial U_3}{\partial x_1}\right)\Pi_1 + \left(M_2^2 - i\omega\frac{\partial U_3}{\partial x_2}\right)\Pi_2\right\}$$

$$c = k^2\left\{ \begin{array}{l} -i\omega\left(\frac{\partial U_1}{\partial x_3}\Pi_{31} + \frac{\partial U_2}{\partial x_3}\Pi_{32}\right)\left[\left(M_1^2 - i\omega\frac{\partial U_3}{\partial x_1}\right)\Pi_1 + \left(M_2^2 - i\omega\frac{\partial U_3}{\partial x_2}\right)\Pi_2\right] \\ \left\{ \begin{array}{l} \Omega^2\left[\omega^2 - (N^2)' + i\omega\frac{\partial U_3}{\partial x_3}\right] \\ + \frac{\partial U_1}{\partial x_3}\frac{\partial U_3}{\partial x_1}\omega\left(\omega + i\frac{U_1}{R} + i\frac{\partial U_2}{\partial x_2}\right) + i\frac{\partial U_2}{\partial x_3}\frac{\partial U_3}{\partial x_1}\omega\left(\bar{F} + \frac{U_2}{R} - \frac{\partial U_1}{\partial x_2}\right) \\ + \frac{\partial U_3}{\partial x_2}\frac{\partial U_2}{\partial x_3}\omega\left(\omega + i\frac{\partial U_1}{\partial x_1}\right) - i\frac{\partial U_1}{\partial x_3}\frac{\partial U_3}{\partial x_2}\omega\left(\bar{F} + \frac{\partial U_2}{\partial x_1}\right) \end{array}\right\} \end{array}\right\} \tag{68}$$


**Step 3.3**: By solving equation (67), we obtain expressions for the vertical wave number and phase function as formula (69). Here $X_3(0)$ is the vertical phase of sea surface.

$$K_3 = \frac{\partial X_3(x_3)}{\partial x_3} = K_{31} \pm K_{32} \qquad\qquad X_3(x_3) = X_3(0) + X_{31}(x_3) \pm X_{32}(x_3) \tag{69}$$

$$K_{31} = -\frac{k}{2\Omega^2}\left\{-i\omega\left(\frac{\partial U_1}{\partial x_3}\Pi_{31} + \frac{\partial U_2}{\partial x_3}\Pi_{32}\right) + \left(M_1^2 - i\omega\frac{\partial U_3}{\partial x_1}\right)\Pi_1 + \left(M_2^2 - i\omega\frac{\partial U_3}{\partial x_2}\right)\Pi_2\right\}$$

$$K_{32} = \frac{1}{\Omega^2}\left\{\Omega^2\left[(N^2)^* - \omega^2\right](\Pi_4)'\right\}^{1/2}k$$

$$X_{31}(x_3) = \int_0^{x_3} K_{31}(x_3)dx_3 = -\frac{1}{2}\int_0^{x_3}\frac{1}{\Omega^2}\left\{-i\omega\left(\frac{\partial U_1}{\partial x_3}\Pi_{31} + \frac{\partial U_2}{\partial x_3}\Pi_{32}\right) + \left(M_1^2 - i\omega\frac{\partial U_3}{\partial x_1}\right)\Pi_1 + \left(M_2^2 - i\omega\frac{\partial U_3}{\partial x_2}\right)\Pi_2\right\}kdx_3$$

$$X_{32}(x_3) = \int_0^{x_3} K_{32}(x_3)dx_3 = \int_0^{x_3}\frac{1}{\Omega^2}\left\{\Omega^2\left[(N^2)^* - \omega^2\right](\Pi_4)'\right\}^{1/2}kdx_3 \tag{70}$$

$$(N^2)^* = (N^2)' - i\omega\frac{\partial U_3}{\partial x_3} - \left\{ \begin{array}{l} \frac{\partial U_1}{\partial x_3}\frac{\partial U_3}{\partial x_1}\frac{\omega\left(\omega + i\frac{U_1}{R} + i\frac{\partial U_2}{\partial x_2}\right)}{\Omega^2} + i\frac{\partial U_2}{\partial x_3}\frac{\partial U_3}{\partial x_1}\frac{\omega\left(\bar{F} + \frac{U_2}{R} - \frac{\partial U_1}{\partial x_2}\right)}{\Omega^2} \\ + \frac{\partial U_3}{\partial x_2}\frac{\partial U_2}{\partial x_3}\frac{\omega\left(\omega + i\frac{\partial U_1}{\partial x_1}\right)}{\Omega^2} - i\frac{\partial U_1}{\partial x_3}\frac{\partial U_3}{\partial x_2}\frac{\omega\left(\bar{F} + \frac{\partial U_2}{\partial x_1}\right)}{\Omega^2} \end{array}\right\}$$
$$+ \frac{1}{4(\Pi_4)'\Omega^2}\left\{ \begin{array}{l} i\omega\left(\frac{\partial U_1}{\partial x_3}\Pi_{31} + \frac{\partial U_2}{\partial x_3}\Pi_{32}\right) \\ + \left(M_1^2 - i\omega\frac{\partial U_3}{\partial x_1}\right)\Pi_1 + \left(M_2^2 - i\omega\frac{\partial U_3}{\partial x_2}\right)\Pi_2 \end{array}\right\}^2$$





**Step 3.4**: According to equation (64), $\overline{\mu}_3$ can be expressed as follows. Without considering the influence of atmospheric pressure and water level, substitute equation (71) into the upper and lower boundary conditions (61) and (62). The results are shown as equations (72)- (73). The solutions of equations (72)-(73) is presented as (74)-(75).

**Step 3.5**: Thus, we obtain the formula of $\overline{\mu}_3$ and $\frac{\partial \overline{\mu}_3}{\partial x_3}$ as (76)-(77).

$$\overline{\mu}_3 = A_1 \exp\{i[X_3(0) + X_{31}(x_3) + X_{32}(x_3)]\} + A_2 \exp\{i[X_3(0) + X_{31}(x_3) - X_{32}(x_3)]\} \tag{71}$$

$$-i\overline{\omega}\eta = A_1 \exp\{i[X_3(0)]\} + A_2 \exp\{i[X_3(0)]\} \tag{72}$$

$$\hat{\delta}_{-H} = A_1 \exp\{i[X_3(0) + X_{31}(-H) + X_{32}(-H)]\} + A_2 \exp\{i[X_3(0) + X_{31}(-H) - X_{32}(-H)]\} \tag{73}$$

$$
\begin{aligned}
A_1 &= \frac{i\overline{\omega}\eta \exp\{i[X_{31}(-H) - X_{32}(-H)]\} + \hat{\delta}_{-H}}{\exp\{i[X_3(0) + X_{31}(-H) + X_{32}(-H)]\} - \exp\{i[X_3(0) + X_{31}(-H) - X_{32}(-H)]\}} \\
&= i\overline{\omega}_0\eta \frac{\exp\{-iX_3(0)\}\exp\{-iX_{32}(-H)\}}{\exp\{iX_{32}(-H)\} - \exp\{-iX_{32}(-H)\}} + \hat{\delta}_{-H}\frac{\exp\{-iX_3(0)\}\exp\{-iX_{31}(-H)\}}{\exp\{iX_{32}(-H)\} - \exp\{-iX_{32}(-H)\}}
\end{aligned}
\tag{74}
$$

$$
\begin{aligned}
A_2 &= \frac{-i\overline{\omega}\eta \exp\{i[X_{31}(-H) + X_{32}(-H)]\} - \hat{\delta}_{-H}}{\exp\{i[X_3(0) + X_{31}(-H) + X_{32}(-H)]\} - \exp\{i[X_3(0) + X_{31}(-H) - X_{32}(-H)]\}} \\
&= -i\overline{\omega}\eta \frac{\exp\{-iX_3(0)\}\exp\{iX_{32}(-H)\}}{\exp\{iX_{32}(-H)\} - \exp\{-iX_{32}(-H)\}} - \hat{\delta}_{-H}\frac{\exp\{-iX_3(0)\}\exp\{-iX_{31}(-H)\}}{\exp\{iX_{32}(-H)\} - \exp\{-iX_{32}(-H)\}}
\end{aligned}
\tag{75}
$$

$$\overline{\mu}_3 = i\overline{\omega}\eta \exp\{iX_{31}(x_3)\}\frac{\sin\{X_{32}(x_3) - X_{32}(-H)\}}{\sin\{X_{32}(-H)\}} + \hat{\delta}_{-H}\exp\{iX_{31}(x_3)\}\exp\{-iX_{31}(-H)\}\frac{\sin\{X_{32}(x_3)\}}{\sin\{X_{32}(-H)\}} \tag{76}$$

$$
\begin{aligned}
\frac{\partial \overline{\mu}_3}{\partial x_3} &= -\overline{\omega}\eta \exp\{iX_{31}(x_3)\}\frac{\sin\{X_{32}(x_3) - X_{32}(-H)\}}{\sin\{X_{32}(-H)\}}k_{31} + i\overline{\omega}\eta \exp\{iX_{31}(x_3)\}\frac{\cos\{X_{32}(x_3) - X_{32}(-H)\}}{\sin\{X_{32}(-H)\}}k_{32} \\
&\quad + i\hat{\delta}_{-H}\exp\{iX_{31}(x_3)\}\exp\{-iX_{31}(-H)\}\frac{\sin\{X_{32}(x_3)\}}{\sin\{X_{32}(-H)\}}k_{31} \\
&\quad + \hat{\delta}_{-H}\exp\{iX_{31}(x_3)\}\exp\{-iX_{31}(-H)\}\frac{\cos\{X_{32}(x_3)\}}{\sin\{X_{32}(-H)\}}k_{32}
\end{aligned}
\tag{77}
$$

**Step 4.1**: Substituting $\overline{\mu}_3$ and $\frac{\partial \overline{\mu}_3}{\partial x_3}$, i.e. (76)-(77), into equation (55), we obtain the formula of $\phi$ as (82).

**Step 4.2**: According to equation (53), we obtain the formula of $\mu_3$ as (80).



**Step 4.3**: Substituting $\overline{\mu}_3$ and $\phi$, i.e. (76) and (82), into equation (45)-(46) and (48), we obtain $\{\mu_1, \mu_2, \beta\}$ as (78)-(79) and (81).

$$
\begin{aligned}
\mu_1 = & -\frac{\overline{\omega}\eta}{(\Pi_4)'}\exp\{iX_{31}(x_3)\}\left\{
\begin{array}{c}
\frac{k_{32}}{k}\left[\Pi_1 + i\Omega^2\Lambda_1\left(\frac{\delta_{-H}}{\omega k}\right)\right]\frac{\cos\{X_{32}(x_3)-X_{32}(-H)\}}{\sin\{X_{32}(-H)\}} \\
+i\left\{\left(\frac{k_{31}}{k}+\Lambda\right)\left[\Pi_1 + i\Omega^2\Lambda_1\left(\frac{\delta_{-H}}{\omega k}\right)\right] + i(\Pi_4)'\Lambda_1\right\}\frac{\sin\{X_{32}(x_3)-X_{32}(-H)\}}{\sin\{X_{32}(-H)\}}
\end{array}
\right\} \\
& -\frac{\widehat{\delta}_{-H}}{(\Pi_4)'}\exp\{iX_{31}(x_3)\}\exp\{-iX_{31}(-H)\}\left\{
\begin{array}{c}
\left\{\left(\frac{k_{31}}{k}+\Lambda\right)\left[\Pi_1 + i\Omega^2\Lambda_1\left(\frac{\delta_{-H}}{\omega k}\right)\right] + i(\Pi_4)'\Lambda_1\right\}\frac{\sin\{X_{32}(x_3)\}}{\sin\{X_{32}(-H)\}} \\
-i\frac{k_{32}}{k}\left[\Pi_1 + i\Omega^2\Lambda_1\left(\frac{\delta_{-H}}{\omega k}\right)\right]\frac{\cos\{X_{32}(x_3)\}}{\sin\{X_{32}(-H)\}}
\end{array}
\right\}
\end{aligned}
\tag{78}
$$

$$
\begin{aligned}
\mu_2 = & \frac{\overline{\omega}\eta}{(\Pi_4)'}\exp\{iX_{31}(x_3)\}\left\{
\begin{array}{c}
\left[(\Pi_4)'\Lambda_2 - i\left(\frac{k_{31}}{k}+\Lambda\right)\Pi_2 + \left(\frac{k_{31}}{k}+\Lambda\right)\Omega^2\Lambda_2\left(\frac{\delta_{-H}}{\omega k}\right)\right]\frac{\sin\{X_{32}(x_3)-X_{32}(-H)\}}{\sin\{X_{32}(-H)\}} \\
-\frac{k_{32}}{k}\left[\Pi_2 + i\Omega^2\Lambda_2\left(\frac{\delta_{-H}}{\omega k}\right)\right]\frac{\cos\{X_{32}(x_3)-X_{32}(-H)\}}{\sin\{X_{32}(-H)\}}
\end{array}
\right\} \\
& +\frac{\widehat{\delta}_{-H}}{(\Pi_4)'}\exp\{iX_{31}(x_3)\}\exp\{-iX_{31}(-H)\}\left\{
\begin{array}{c}
i\frac{k_{32}}{k}\left[\Pi_2 + i\Omega^2\Lambda_2\left(\frac{\delta_{-H}}{\omega k}\right)\right]\frac{\cos\{X_{32}(x_3)\}}{\sin\{X_{32}(-H)\}} \\
-i\left\{(\Pi_4)'\Lambda_2 - i\left(\frac{k_{31}}{k}+\Lambda\right)\left[\Pi_2 + i\Omega^2\Lambda_2\left(\frac{\delta_{-H}}{\omega k}\right)\right]\right\}\frac{\sin\{X_{32}(x_3)\}}{\sin\{X_{32}(-H)\}}
\end{array}
\right\}
\end{aligned}
\tag{79}
$$

$$
\begin{aligned}
\mu_3 = & i\frac{\overline{\omega}\eta}{(\Pi_4)'}\exp\{iX_{31}(x_3)\}\left\{
\begin{array}{c}
\left[(\Pi_4)' + \Omega^2\left(\frac{k_{31}}{k}+\Lambda\right)\left(\frac{\delta_{-H}}{\omega k}\right)\right]\frac{\sin\{X_{32}(x_3)-X_{32}(-H)\}}{\sin\{X_{32}(-H)\}} \\
-i\Omega^2\frac{k_{32}}{k}\left(\frac{\delta_{-H}}{\omega k}\right)\frac{\cos\{X_{32}(x_3)-X_{32}(-H)\}}{\sin\{X_{32}(-H)\}}
\end{array}
\right\} \\
& +\frac{\widehat{\delta}_{-H}}{(\Pi_4)'}\exp\{iX_{31}(x_3)\}\exp\{-iX_{31}(-H)\}\left\{
\begin{array}{c}
\left[(\Pi_4)' + \Omega^2\left(\frac{k_{31}}{k}+\Lambda\right)\left(\frac{\delta_{-H}}{\omega k}\right)\right]\frac{\sin\{X_{32}(x_3)\}}{\sin\{X_{32}(-H)\}} \\
-i\Omega^2\frac{k_{32}}{k}\left(\frac{\delta_{-H}}{\omega k}\right)\frac{\cos\{X_{32}(x_3)\}}{\sin\{X_{32}(-H)\}}
\end{array}
\right\}
\end{aligned}
\tag{80}
$$

$$
\begin{aligned}
\beta = & -\frac{\overline{\omega}\eta}{(\Pi_4)'}\left(\frac{\Omega^2}{g\omega}\right)\exp\{iX_{31}(x_3)\}\left\{
\begin{array}{c}
\left\{\frac{(N^2)'}{\Omega^2}(\Pi_4)' - \left(\frac{k_{31}}{k}+\Lambda\right)\right\}\left[\frac{M_1^2}{\Omega^2}\Pi_1 + \frac{M_2^2}{\Omega^2}\Pi_2 - (N^2)'\left(\frac{\delta_{-H}}{\omega k}\right)\right]\frac{\sin\{X_{32}(x_3)-X_{32}(-H)\}}{\sin\{X_{32}(-H)\}} \\
+i\frac{k_{32}}{k}\left[\frac{M_1^2}{\Omega^2}\Pi_1 + \frac{M_2^2}{\Omega^2}\Pi_2 - (N^2)'\left(\frac{\delta_{-H}}{\omega k}\right)\right]\frac{\cos\{X_{32}(x_3)-X_{32}(-H)\}}{\sin\{X_{32}(-H)\}}
\end{array}
\right\} \\
& -\frac{\widehat{\delta}_{-H}}{(\Pi_4)'}\exp\{iX_{31}(x_3)\}\exp\{-iX_{31}(-H)\}\left(\frac{\Omega^2}{g\omega}\right)\left\{
\begin{array}{c}
\frac{k_{32}}{k}\left[\frac{M_1^2}{\Omega^2}\Pi_1 + \frac{M_2^2}{\Omega^2}\Pi_2 - (N^2)'\left(\frac{\delta_{-H}}{\omega k}\right)\right]\frac{\cos\{X_{32}(x_3)\}}{\sin\{X_{32}(-H)\}} \\
+i\left[\left(\frac{k_{31}}{k}+\Lambda\right)\left[\frac{M_1^2}{\Omega^2}\Pi_1 + \frac{M_2^2}{\Omega^2}\Pi_2 - (N^2)'\left(\frac{\delta_{-H}}{\omega k}\right)\right] - \frac{(N^2)'}{\Omega^2}(\Pi_4)'\right]\frac{\sin\{X_{32}(x_3)\}}{\sin\{X_{32}(-H)\}}
\end{array}
\right\}
\end{aligned}
\tag{81}
$$





$$\phi = -\frac{\overline{\omega}\eta}{(\Pi_4)'}\frac{\Omega^2}{\omega k}\exp\{iX_{31}(x_3)\}\left\{\begin{array}{c}\dfrac{k_{32}}{k}\dfrac{\cos\{X_{32}(x_3)-X_{32}(-H)\}}{\sin\{X_{32}(-H)\}}\\[1.5em]+i\left(\dfrac{k_{31}}{k}+\Lambda\right)\dfrac{\sin\{X_{32}(x_3)-X_{32}(-H)\}}{\sin\{X_{32}(-H)\}}\end{array}\right\}$$
$$-\frac{\hat{\delta}_{-H}}{(\Pi_4)'}\frac{\Omega^2}{\omega k}\exp\{iX_{31}(x_3)\}\exp\{-iX_{31}(-H)\}\left\{\begin{array}{c}\left(\dfrac{k_{31}}{k}+\Lambda\right)\dfrac{\sin\{X_{32}(x_3)\}}{\sin\{X_{32}(-H)\}}\\[1.5em]-i\dfrac{k_{32}}{k}\dfrac{\cos\{X_{32}(x_3)\}}{\sin\{X_{32}(-H)\}}\end{array}\right\} \tag{82}$$

$$\Lambda = -\left\{i\frac{\omega}{\Omega^2}\left(\frac{\partial U_1}{\partial x_3}\Pi_{31}+\frac{\partial U_2}{\partial x_3}\Pi_{32}\right)+\left\{\begin{array}{c}\left[\omega^2-(N^2)'+i\omega\dfrac{\partial U_3}{\partial x_3}\right]\\[1em]+\dfrac{\partial U_1}{\partial x_3}\dfrac{\partial U_3}{\partial x_1}\dfrac{\omega\left(\omega+i\dfrac{U_1}{R}+i\dfrac{\partial U_2}{\partial x_2}\right)}{\Omega^2}+i\dfrac{\partial U_2}{\partial x_3}\dfrac{\partial U_3}{\partial x_1}\dfrac{\omega\left(\overline{F}+\dfrac{U_2}{R}-\dfrac{\partial U_1}{\partial x_2}\right)}{\Omega^2}\\[1em]+\dfrac{\partial U_3}{\partial x_2}\dfrac{\partial U_2}{\partial x_3}\dfrac{\omega\left(\omega+i\dfrac{\partial U_1}{\partial x_1}\right)}{\Omega^2}-i\dfrac{\partial U_1}{\partial x_3}\dfrac{\partial U_3}{\partial x_2}\dfrac{\omega\left(\overline{F}+\dfrac{\partial U_2}{\partial x_1}\right)}{\Omega^2}\end{array}\right\}\left(\dfrac{\delta_{-H}}{\omega k}\right)\right\} \tag{83}$$

$$\Lambda_1 = \frac{1}{\Omega^2}\left[\frac{\partial U_1}{\partial x_3}\left(\omega+i\frac{U_1}{R}+i\frac{\partial U_2}{\partial x_2}\right)+i\frac{\partial U_2}{\partial x_3}\left(\overline{F}+\frac{U_2}{R}-\frac{\partial U_1}{\partial x_2}\right)\right]$$

$$\Lambda_2 = \frac{1}{\Omega^2}\left[\frac{\partial U_2}{\partial x_3}\left(\omega+i\frac{\partial U_1}{\partial x_1}\right)-i\frac{\partial U_1}{\partial x_3}\left(\overline{F}+\frac{\partial U_2}{\partial x_1}\right)\right]$$


The surface complex frequency dispersion relation, i.g. equation (84), can be derived by substituting $\phi$ equation (82) into the boundary condition (60), neglecting the influence of $\hat{\delta}_{-H}$. According to the equation (85), waves exhibit distinct motion properties across three frequency segments, namely ocean waves, internal waves, and inertial waves.

$$\left(\frac{\overline{\rho}}{\rho_0}\right)_{x_3=0}\left(\frac{\omega}{\overline{\omega}}\right)(\Pi_4)'(gk)+\Omega_0^2\left\{\frac{(k_{32})_0}{k}\frac{\cos\{X_{32}(-H)\}}{\sin\{X_{32}(-H)\}}-i\left[\frac{(k_{31})_0}{k}+\Lambda\right]\right\}=0 \tag{84}$$

$$[\text{Re}\{\omega\}]_{SeaWaves} > \text{Re}\left\{\left[(N^2)^*\right]^{1/2}\right\} > [\text{Re}\{\omega\}]_{InternalWaves} > \left[\left(\overline{F}+\frac{\partial U_2}{\partial x_1}\right)\left(\overline{F}+\frac{U_2}{R}-\frac{\partial U_1}{\partial x_2}\right)\right]^{1/2} > [\text{Re}\{\omega\}]_{InertialWaves} \tag{85}$$

### 190  2.1.2 Solutions for ocean surface waves

For ocean surface waves, according to equations (69)-(70), $K_3$ and $X_3(x_3)$ are as follows. By substituting $X_{31}(x_3)$ and $X_{32}(x_3)$ into $\{\mu_1,\mu_2,\mu_3,\beta,\phi\}$, i.g. equations (78)-(82), and retaining the real part of the solution, we obtain the solution for ocean surface waves, as manifested in equations (87)-(91). Here, $\alpha=1,2$.

$$K_{31}=0 \qquad\qquad\qquad X_{31}(x_3)=\int_0^{x_3}K_{31}(x_3)dx_3=0 \tag{86}$$





$$K_{32} = \frac{1}{\Omega^2}\left\{\Omega^2[(N^2)^* - \omega^2](\Pi_4)'\right\}^{1/2} k = ik \qquad\qquad X_{32}(x_3) = \int_0^{x_3} K_{32}(x_3)dx_3 = ikx_3$$


$$\mu_1 = \overline{\omega}\eta \left\{ \begin{array}{c} \Pi_1 \dfrac{\cosh\{k(x_3+H)\}}{\sinh\{kH\}} \\[4pt] -\Lambda_1\left[1+\Lambda\Omega^2\left(\dfrac{\delta_{-H}}{\omega k}\right)\right]\dfrac{\sinh\{k(x_3+H)\}}{\sinh\{kH\}} \end{array} \right\} - \mu_{(H_{SM})}k_\alpha(U_\alpha)_{x_3=-H}\left\{ \begin{array}{c} \Pi_1 \dfrac{\cosh\{kx_3\}}{\sinh\{kH\}} \\[4pt] -\Lambda_1\left[1+\Lambda\Omega^2\left(\dfrac{\delta_{-H}}{\omega k}\right)\right]\dfrac{\sinh\{kx_3\}}{\sinh\{kH\}} \end{array} \right\} \qquad (87)$$

$$\mu_2 = \overline{\omega}\eta \left\{ \begin{array}{c} \Pi_2 \dfrac{\cosh\{k(x_3+H)\}}{\sinh\{kH\}} \\[4pt] -\Lambda_2\left[1+\Lambda\Omega^2\left(\dfrac{\delta_{-H}}{\omega k}\right)\right]\dfrac{\sinh\{k(x_3+H)\}}{\sinh\{kH\}} \end{array} \right\} - \mu_{(H_{SM})}k_\alpha(U_\alpha)_{x_3=-H}\left\{ \begin{array}{c} \Pi_2 \dfrac{\cosh\{kx_3\}}{\sinh\{kH\}} \\[4pt] -\Lambda_2\left[1+\Lambda\Omega^2\left(\dfrac{\delta_{-H}}{\omega k}\right)\right]\dfrac{\sinh\{kx_3\}}{\sinh\{kH\}} \end{array} \right\} \qquad (88)$$

$$\mu_3 = i\left\{-\overline{\omega}\eta\left[1+\Omega^2\Lambda\left(\frac{\delta_{-H}}{\omega k}\right)\right]\frac{\sinh\{k(x_3+H)\}}{\sinh\{kH\}} + \mu_{(H_{SM})}k_\alpha(U_\alpha)_{x_3=-H}\left[1+\Omega^2\Lambda\left(\frac{\delta_{-H}}{\omega k}\right)\right]\frac{\sinh\{kx_3\}}{\sinh\{kH\}}\right\} \qquad (89)$$

$$\beta = \eta\frac{(N^2)'}{g}\left\{\left[1+\Lambda\Omega^2\left(\frac{\delta_{-H}}{\omega k}\right)\right]\frac{\sinh\{k(x_3+H)\}}{\sinh\{kH\}}\right\} - \mu_{(H_{SM})}k_\alpha(U_\alpha)_{x_3=-H}\frac{(N^2)'}{g\omega}\left\{\left[1+\Lambda\Omega^2\left(\frac{\delta_{-H}}{\omega k}\right)\right]\frac{\sinh\{kx_3\}}{\sinh\{kH\}}\right\} \qquad (90)$$

$$\phi = \eta\frac{\Omega^2}{k}\frac{\cosh\{k(x_3+H)\}}{\sinh\{kH\}} - \mu_{(H_{SM})}k_\alpha(U_\alpha)_{x_3=-H}\frac{\Omega^2}{\omega k}\frac{\cosh\{kx_3\}}{\sinh\{kH\}} \qquad (91)$$

## 2.2 Multi-beam terrain data

In the solution for ocean surface waves, $\mu_{(H_{SM})}$ is a crucial parameter associated with topographic fluctuations (refer to equation (34)). Through proper transformation, we can obtain more information from the bottom spectrum of a bottom

elevation map (Ardhuin and Magne, 2007). In this section, multi-beam terrain data (as depicted in Figure 1a) from the sea area near Qingdao is employed to assess this parameter $\mu_{(H_{SM})}$. The observed region spans longitude (120.4180ºE, 120.4285ºE), latitude (36.0275ºN, 36.0495ºN). The spatial resolution is 1m × 1m. According to the double Fourier series formula (92), the multi-bean terrain data are decomposed into four modes, as illustrated in Figure 2. Here, number of decomposition term is 100, i.g. $(m, n = 0, 1, 2, \cdots, 100)$. To verify the accuracy of the Fourier decomposition, the four

modes are superimposed, and the restoration diagram is presented in Figure 1b.

$$f(x,y) \sim \sum_{m,n=0}^{\infty} \lambda_{m,n}\left[a_{m,n}\cos\frac{\pi m x}{l}\cos\frac{\pi n y}{h} + b_{m,n}\sin\frac{\pi m x}{l}\cos\frac{\pi n y}{h} + c_{m,n}\cos\frac{\pi m x}{l}\sin\frac{\pi n y}{h} + d_{m,n}\sin\frac{\pi m x}{l}\sin\frac{\pi n y}{h}\right] \qquad (92)$$



$$\lambda_{m,n} = \begin{cases} \dfrac{1}{4} & (m = n = 0) \\ \dfrac{1}{2} & (m > 0, n = 0 \text{ 或 } m = 0, n > 0) \\ 1 & (m > 0, n > 0) \end{cases} \qquad (m, n = 0, 1, 2, \cdots)$$

$$a_{m,n} = \frac{1}{lh} \iint\limits_{R} f(x,y) \cos\frac{m\pi x}{l} \cos\frac{n\pi y}{h} dx dy \qquad b_{m,n} = \frac{1}{lh} \iint\limits_{R} f(x,y) \sin\frac{m\pi x}{l} \cos\frac{n\pi y}{h} dx dy$$

$$c_{m,n} = \frac{1}{lh} \iint\limits_{R} f(x,y) \cos\frac{m\pi x}{l} \sin\frac{n\pi y}{h} dx dy \qquad d_{m,n} = \frac{1}{lh} \iint\limits_{R} f(x,y) \sin\frac{m\pi x}{l} \sin\frac{n\pi y}{h} dx dy$$

**Figure 1: Original diagram and restoration diagram for multi-beam terrain data. The color illustrates water depth, denoted in meters.**

The wavelength of ocean surface waves spans approximately 9 meters to 900 meters. According to Figure 2, the topographic Fourier coefficient $\mu_{(H_{SM})}$ corresponding to ocean surface waves ranges from 0.1 meters to 0.5 meters. Analysis related to this parameter is given in section 3.2.





**Figure 2: Fourier coefficients $\{a, b, c, d\}$ for four modes. The horizontal and vertical coordinates denote the wavelength.**

## 3 Results

For ocean surface waves, formula of $\{\overline{\omega}, \Omega^2, \Pi_1, \Pi_2, \Lambda_1, \Lambda_2\}$ can be rewritten as follows. Substitute $\{\overline{\omega}, \Omega^2, \Pi_1, \Pi_2, \Lambda_1, \Lambda_2\}$ into equations (87)-(89) of $\{\mu_1, \mu_2, \mu_3\}$, and omit the term of topographic slope $\delta_{-H}$, we obtain the amplitudes of orbital velocities for monochromatic ocean surface wave considering topographic fluctuations and vertical shear of current as equations (94)-(96).

*For gravity waves*                                          *For ocean surface waves*

$$\overline{\omega} = \omega\left[1 + igk\left(\frac{\overline{\rho}}{\rho_0}\right)_{x_3=0}\left(\frac{\delta_{-H}}{\omega k}\right)\right] \qquad\qquad \overline{\omega} = \omega \qquad\qquad (93)$$



$$\Omega^2 = \left(\omega + i\frac{\partial U_1}{\partial x_1}\right)\left(\omega + i\frac{U_1}{R} + i\frac{\partial U_2}{\partial x_2}\right) - \left(\overline{F} + \frac{\partial U_2}{\partial x_1}\right)\left(\overline{F} + \frac{U_2}{R} - \frac{\partial U_1}{\partial x_2}\right) \qquad \Omega^2 = \omega^2$$

$$\Pi_1 = \frac{k_1}{k}\frac{\omega + i\frac{U_1}{R} + i\frac{\partial U_2}{\partial x_2}}{\omega} + i\frac{k_2}{k}\frac{\overline{F} + \frac{U_2}{R} - \frac{\partial U_1}{\partial x_2}}{\omega} \qquad \Pi_1 = \frac{k_1}{k}$$

$$\Pi_2 = -i\frac{k_1}{k}\frac{\overline{F} + \frac{\partial U_2}{\partial x_1}}{\omega} + \frac{k_2}{k}\frac{\omega + i\frac{\partial U_1}{\partial x_1}}{\omega} \qquad \Pi_2 = \frac{k_2}{k}$$

$$\Lambda_1 = \frac{1}{\Omega^2}\left[\frac{\partial U_1}{\partial x_3}\left(\omega + i\frac{U_1}{R} + i\frac{\partial U_2}{\partial x_2}\right) + i\frac{\partial U_2}{\partial x_3}\left(\overline{F} + \frac{U_2}{R} - \frac{\partial U_1}{\partial x_2}\right)\right] \qquad \Lambda_1 = \frac{1}{\omega}\frac{\partial U_1}{\partial x_3}$$

$$\Lambda_2 = \frac{1}{\Omega^2}\left[\frac{\partial U_2}{\partial x_3}\left(\omega + i\frac{\partial U_1}{\partial x_1}\right) - i\frac{\partial U_1}{\partial x_3}\left(\overline{F} + \frac{\partial U_2}{\partial x_1}\right)\right] \qquad \Lambda_2 = \frac{1}{\omega}\frac{\partial U_2}{\partial x_3}$$

$$\mu_1 = \omega\eta\left\{\frac{k_1}{k}\frac{\cosh\{k(x_3+H)\}}{\sinh\{kH\}} - \frac{1}{\omega}\frac{\partial U_1}{\partial x_3}\frac{\sinh\{k(x_3+H)\}}{\sinh\{kH\}}\right\} - \mu_{(H_{SM})}k_\alpha(U_\alpha)_{x_3=-H}\left\{\frac{k_1}{k}\frac{\cosh\{kx_3\}}{\sinh\{kH\}} - \frac{1}{\omega}\frac{\partial U_1}{\partial x_3}\frac{\sinh\{kx_3\}}{\sinh\{kH\}}\right\} \qquad (94)$$

$$\mu_2 = \omega\eta\left\{\frac{k_2}{k}\frac{\cosh\{k(x_3+H)\}}{\sinh\{kH\}} - \frac{1}{\omega}\frac{\partial U_2}{\partial x_3}\frac{\sinh\{k(x_3+H)\}}{\sinh\{kH\}}\right\} - \mu_{(H_{SM})}k_\alpha(U_\alpha)_{x_3=-H}\left\{\frac{k_2}{k}\frac{\cosh\{kx_3\}}{\sinh\{kH\}} - \frac{1}{\omega}\frac{\partial U_2}{\partial x_3}\frac{\sinh\{kx_3\}}{\sinh\{kH\}}\right\} \qquad (95)$$

$$\mu_3 = i\left\{-\omega\eta\frac{\sinh\{k(x_3+H)\}}{\sinh\{kH\}} + \mu_{(H_{SM})}k_\alpha(U_\alpha)_{x_3=-H}\frac{\sinh\{kx_3\}}{\sinh\{kH\}}\right\} \qquad (96)$$

in which, terms related to $\left\{\frac{\partial U_1}{\partial x_3}, \frac{\partial U_2}{\partial x_3}\right\}$ are vertical shear of current, and terms of $\mu_{(H_{SM})}k_\alpha(U_\alpha)_{x_3=-H}$ are topographic fluctuations.

In deep ocean, the influence of seafloor conditions on ocean surface waves is minimal, whereas topographic fluctuations exhibit a significant role in shallow waters. Next, we present an analysis of the variations in velocity amplitudes and wave particle trajectory with depth in deep and shallow waters. The solutions of the unified linearized wave, whether obtained through Fourier integrals as in Section 2.1 or Fourier series, exhibit identical mathematical form. The impact of topographic fluctuations and vertical shear of background current on ocean surface waves is examined below, focusing on a single characteristic wave.

### 3.1 In deep ocean

In deep ocean, the amplitudes $\{\mu_1, \mu_2, \mu_3\}$ of orbital velocities for ocean surface waves and their variations $\{\Delta\mu_1, \Delta\mu_2, \Delta\mu_3\}$ due to vertical shear of current are expressed as equations (97)-(99), omitting terms of $\mu_{(H_{SM})}k_\alpha(U_\alpha)_{x_3=-H}$. In this section, results for deep ocean are given, and variable relationship formulas $\left\{\omega^2 = gk, \frac{2\pi}{L} = k, \frac{2\pi}{T} = \omega\right\}$ are employed here. We



posits that ocean surface waves propagate towards the east, thus $k_1 = k, k_2 = 0$. Water depth $H$ is 1000m, wave amplitude $\eta$ is 1m, vertical shear of current $\frac{\partial U_1}{\partial x_3} = \{0.1\text{s}^{-1}, 0.2\text{s}^{-1}, -0.2\text{s}^{-1}\}, \frac{\partial U_2}{\partial x_3} = 0$.

$$\mu_1 = \omega\eta\left\{\frac{k_1}{k}\frac{\cosh\{k(x_3+H)\}}{\sinh\{kH\}} - \frac{1}{\omega}\frac{\partial U_1}{\partial x_3}\frac{\sinh\{k(x_3+H)\}}{\sinh\{kH\}}\right\} \qquad \Delta\mu_1 = -\eta\frac{\partial U_1}{\partial x_3}\frac{\sinh\{k(x_3+H)\}}{\sinh\{kH\}} \qquad (97)$$

$$\mu_2 = \omega\eta\left\{\frac{k_2}{k}\frac{\cosh\{k(x_3+H)\}}{\sinh\{kH\}} - \frac{1}{\omega}\frac{\partial U_2}{\partial x_3}\frac{\sinh\{k(x_3+H)\}}{\sinh\{kH\}}\right\} \qquad \Delta\mu_2 = -\eta\frac{\partial U_2}{\partial x_3}\frac{\sinh\{k(x_3+H)\}}{\sinh\{kH\}} \qquad (98)$$

$$\mu_3 = i\left\{-\omega\eta\frac{\sinh\{k(x_3+H)\}}{\sinh\{kH\}}\right\} \qquad \Delta\mu_3 = 0 \qquad (99)$$

Evaluation of $\mu_1$ and $\Delta\mu_1$ for different wavelengths are shown in Figure 3. According to Figure 3, the amplitude of ocean
surface wave velocity gradually decreases with depth, approaching zero when the depth exceeds 100 meters. The larger the
flow shear $\frac{\partial U_1}{\partial x_3}$, the larger the variation in velocity amplitude for ocean surface waves. The depth of influence of flow shear
increases proportionally with the longer wavelength. In addition, when the vertical shear of current is negative, it enhances
the influence of flow shear ($\Delta\mu_1 > 0$), while when the vertical shear of current is positive, there is a decrease in velocity
amplitude ($\Delta\mu_1 < 0$), indicating a weakening effect. The black line in Figure 3b represents the classical solution of
amplitude $\mu_1$, while the three red lines depict amplitude $\mu_1$ considering different flow shear $\{0.1\text{s}^{-1}, 0.2\text{s}^{-1}, -0.2\text{s}^{-1}\}$ for
ocean surface waves with a wavelength of 200m.

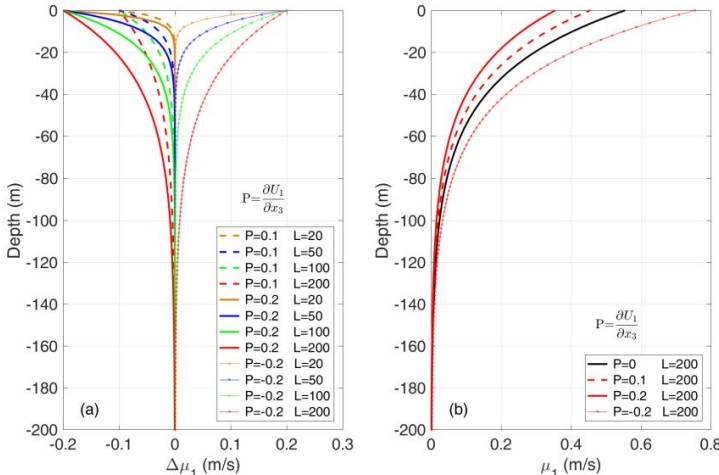

**Figure 3: Amplitude of velocity for ocean surface waves considering vertical shear of current.**

The velocity components in the $x$-axis and $z$-axis $\{u, w\}$, as well as the wave particle trajectory for ocean surface waves, can
be described by equation (100) for a single characteristic wave propagating eastward. The vertical shear of current influences



the horizontal radius of wave particle trajectory, denote as $Rx$, with variation of $\Delta rx = -\left(\frac{1}{\omega}\right)\eta\frac{\partial U_1}{\partial x_3}\frac{\sinh\{k(x_3+H)\}}{\sinh\{kH\}}$, while has no

effect on the vertical radius $Rz$.

$$u = \mu_1 \cos(kx - \omega t) \qquad\qquad w = \mu_3 \sin(kx - \omega t)$$

$$\frac{(x - x_0)^2}{\left[\eta\frac{\cosh\{k(x_3 + H)\}}{\sinh\{kH\}} - \left(\frac{1}{\omega}\right)\eta\frac{\partial U_1}{\partial x_3}\frac{\sinh\{k(x_3 + H)\}}{\sinh\{kH\}}\right]^2} + \frac{(z - z_0)^2}{\left[\eta\frac{\sinh\{k(x_3 + H)\}}{\sinh\{kH\}}\right]^2} = 1 \qquad (100)$$

Evaluation of $Rx$ and $\Delta rx$ for different wavelengths are shown in Figure 4. The impact of vertical current shear on the
horizontal radius is most pronounced at the ocean surface, gradually diminishing with increasing depth. The influence depth

on ocean surface waves with a wavelength of 200m is approximately 100m, whereas for ocean surface waves with a
wavelength of 20m, it is less than 20m. The black line in Figure 4b represents the classical solution for the radius of the
wave particle trajectory, while the three red lines depict the radii of ocean surface waves with a wavelength of 200m,
influenced by varying shear of current. Among them, the red dashed and red solid lines denote $\frac{\partial U_1}{\partial x_3} = \{0.1s^{-1}, 0.2s^{-1}\}$
respectively, while the red dot line represents $\frac{\partial U_1}{\partial x_3} = -0.2s^{-1}$. For simplicity, here the vertical profile of $U_1$ is not considered

(hereinafter the same). Therefore, the positive vertical shear of the current weakens the ocean surface waves ($\Delta rx < 0$),
while the negative one strengthens the ocean surface waves ($\Delta rx > 0$). The aforementioned characteristics are applicable to
ocean surface waves with different wavelengths $\{20m, 50m, 100m\}$. Although the water depth $H$ is 1000m, the modification
$\Delta rx$ becomes negligible beyond a depth of 100m; hence, this comparison is not described for depths exceeding 100m here.

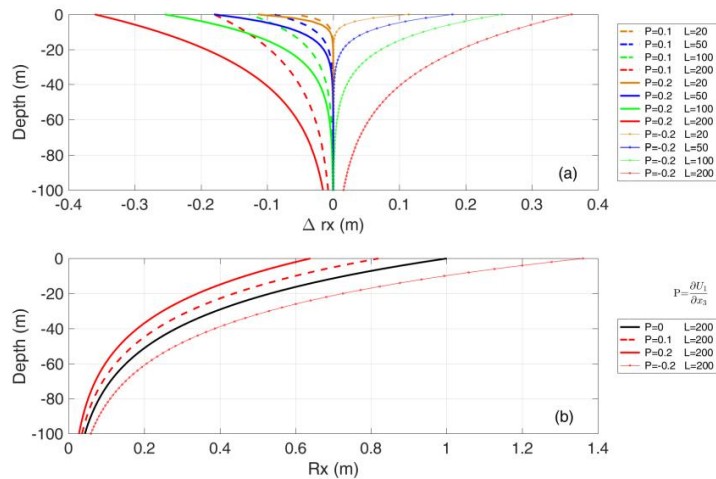

**Figure 4: The horizontal radius for wave particle trajectory considering vertical shear of current.**




Vertical structure of motion trajectory of one wave particle with a wavelength of 200m is shown in Figure 5. The classical solution of wave particle trajectory is represented by the black circle, while the red circle represents the wave particle trajectory considering vertical shear of background current. The numerical values denoted by different colors correspond to the horizontal and vertical radii $\{rx, rz\}$ of the black and red circles. Vertical radii of black and red circles are the same, while horizontal radius become smaller or larger in presence of current vertical shear $\frac{\partial U_1}{\partial x_3} = 0.2\text{s}^{-1}$ or $\frac{\partial U_1}{\partial x_3} = -0.2\text{s}^{-1}$. At different depths $\{0m, 20m, 40m, 60m, 80m, 100m\}$, variations of horizontal radius are approximately $\{\pm 0.36m, \pm 0.20m, \pm 0.10m, \pm 0.05m, \pm 0.03m, \pm 0.01m\}$. The remarkable influence of current shear on ocean surface waves, whether strengthening or weakening, is clearly demonstrated by the superposition of black and red circles. Although the water depth $H$ is 1000m, the discrepancy between black and red circles becomes negligible beyond a depth of 100m; hence, this comparison is not depicted for depths exceeding 100m here.

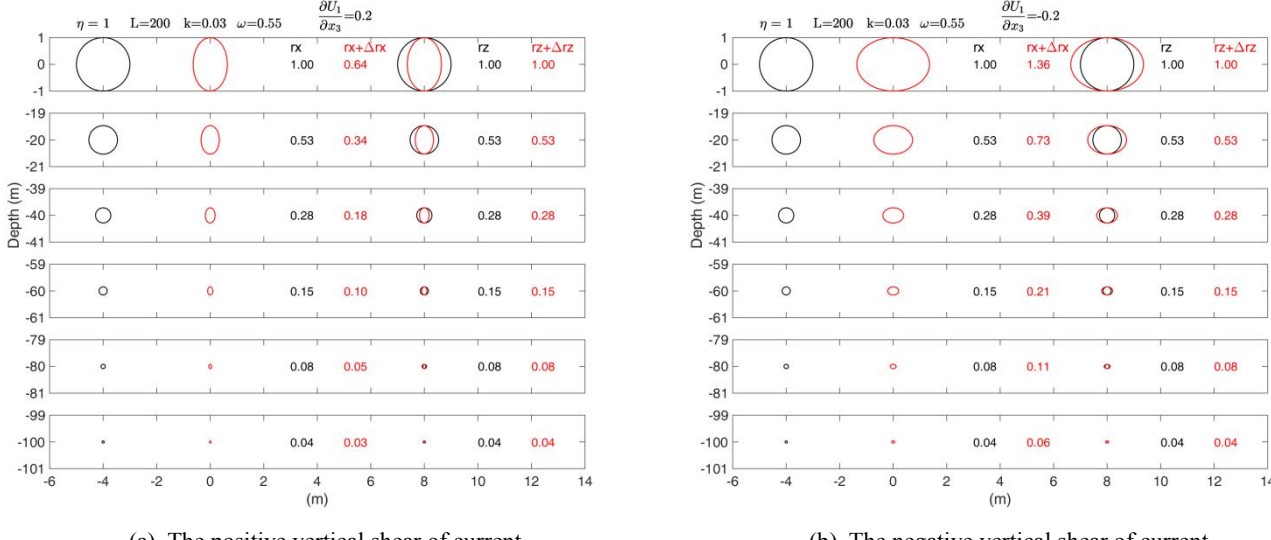

(a) The positive vertical shear of current          (b) The negative vertical shear of current

**Figure 5: Vertical structure of motion trajectory of one wave particle considering vertical shear of current.**

### 3.2 In shallow waters

In shallow waters, it is essential to consider not only the vertical shear of background current, but also the impact of topographic fluctuations on ocean surface waves. In shallow waters, the amplitudes $\{\mu_1, \mu_2, \mu_3\}$ of velocities for ocean surface waves and their variations $\{\Delta\mu_1, \Delta\mu_2, \Delta\mu_3\}$ due to topographic fluctuations are represent as equations (101)-(103), where $\mu_{(H_{SM})}k_\alpha(U_\alpha)_{x_3=-H} = \mu_{(H_{SM})}k_1(U_1)_{x_3=-H} + \mu_{(H_{SM})}k_2(U_2)_{x_3=-H}$, and omitting terms related to $\left\{\frac{\partial U_1}{\partial x_3}, \frac{\partial U_2}{\partial x_3}\right\}$. In this section, variable relationship formulas $\left\{\omega^2 = gk\tanh(kH), \frac{2\pi}{L} = k, \frac{2\pi}{T} = \omega\right\}$ are employed. Similar to the preceding section, it it assumed that ocean surface waves propagate in an eastward direction, i.e. $k_1 = k, k_2 = 0$. Thus, $\mu_{(H_{SM})}k_\alpha(U_\alpha)_{x_3=-H} = \mu_{(H_{SM})}k_1(U_1)_{x_3=-H}$. Water depth $H$ is $\{10m, 30m\}$, wave amplitude $\eta$ is 1m, the topographic Fourier coefficient $\mu_{(H_{SM})} =$



$\{0.1m, 0.5m\}$, current velocities at the ocean bottom $U_1 = \{2\,\mathrm{m/s}, -2\,\mathrm{m/s}\}$, $U_2 = 0$, vertical shear of background current $\left\{P = \frac{\partial U_1}{\partial x_3}, \frac{\partial U_2}{\partial x_3}\right\} = 0\,s^{-1}$.

$$\mu_1 = \omega\eta\left\{\frac{k_1}{k}\frac{\cosh\{k(x_3+H)\}}{\sinh\{kH\}}\right\} - \mu_{(H_{SM})}k_\alpha(U_\alpha)_{x_3=-H}\left\{\frac{k_1}{k}\frac{\cosh\{kx_3\}}{\sinh\{kH\}}\right\} \qquad \Delta\mu_1 = -\mu_{(H_{SM})}k_\alpha(U_\alpha)_{x_3=-H}\left\{\frac{k_1}{k}\frac{\cosh\{kx_3\}}{\sinh\{kH\}}\right\} \qquad (101)$$

$$\mu_2 = \omega\eta\left\{\frac{k_2}{k}\frac{\cosh\{k(x_3+H)\}}{\sinh\{kH\}}\right\} - \mu_{(H_{SM})}k_\alpha(U_\alpha)_{x_3=-H}\left\{\frac{k_2}{k}\frac{\cosh\{kx_3\}}{\sinh\{kH\}}\right\} \qquad \Delta\mu_2 = -\mu_{(H_{SM})}k_\alpha(U_\alpha)_{x_3=-H}\left\{\frac{k_2}{k}\frac{\cosh\{kx_3\}}{\sinh\{kH\}}\right\} \qquad (102)$$

$$\mu_3 = i\left\{-\omega\eta\frac{\sinh\{k(x_3+H)\}}{\sinh\{kH\}} + \mu_{(H_{SM})}k_\alpha(U_\alpha)_{x_3=-H}\frac{\sinh\{kx_3\}}{\sinh\{kH\}}\right\} \qquad \Delta\mu_3 = \mu_{(H_{SM})}k_\alpha(U_\alpha)_{x_3=-H}\frac{\sinh\{kx_3\}}{\sinh\{kH\}} \qquad (103)$$

Evaluation of $\{\mu_1, \mu_3, \Delta\mu_1, \Delta\mu_3\}$ for different wavelengths are shown in Figure 6. Topographic fluctuations exert an influence on the horizontal amplitude of wave particle velocity at both the ocean bottom and surface, while they affect the vertical amplitude of wave particle velocity mainly at the ocean bottom. At the ocean surface, there is a positive correlation between

wavelength and variation of horizontal amplitude of wave particle velocity $\Delta\mu_1$, with longer wavelengths exhibiting greater variations; conversely, at the ocean bottom, there is an inverse relationship between wavelength and variation $\Delta\mu_1$, where shorter wavelengths correspond to greater variation. In Figure 6 (a3, a4, b3, b4), the three lines represents the velocity amplitudes of ocean waves with a wavelength of 50m. The black solid line represents the classical solution of $\mu_1$ or $\mu_3$. The blue solid line represents the $\mu_1$ or $\mu_3$, considering topographic fluctuations (with coefficient $\mu_{(H_{SM})} = 0.5m$) and

background current at the ocean bottom $\{U_1 = 2\,\mathrm{m/s}, U_2 = 0\}$, while the pink dot line represents the $\mu_1$ or $\mu_3$, considering topographic fluctuations (with coefficient $\mu_{(H_{SM})} = 0.5m$) and background current at the ocean bottom $\{U_1 = -2\,\mathrm{m/s}, U_2 = 0\}$. The direction of background current at the ocean bottom influences whether the impact on ocean surface waves is attenuating or amplifying.

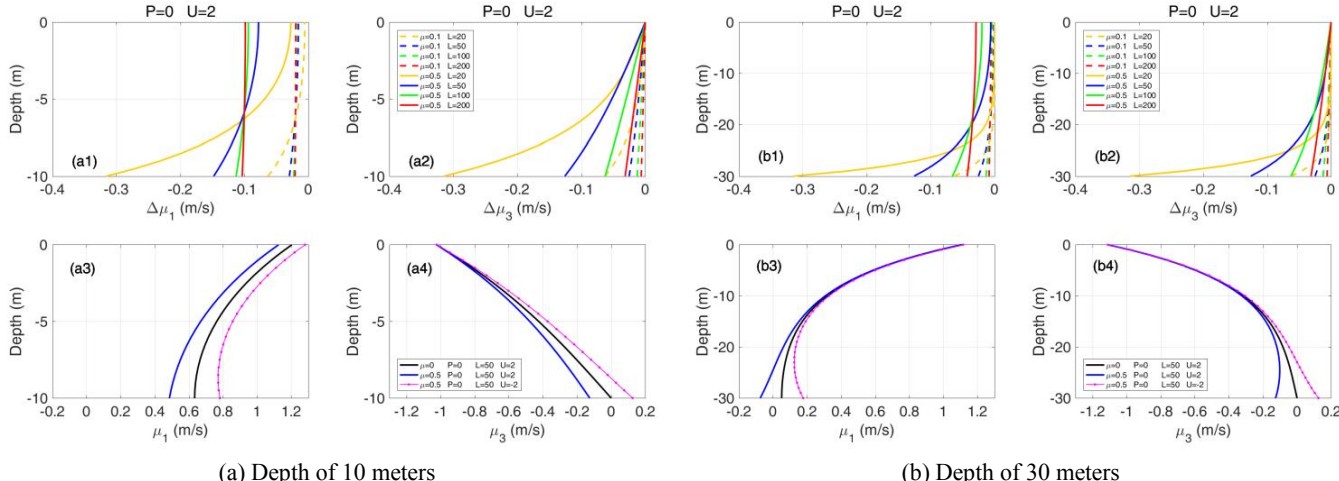

(a) Depth of 10 meters                        (b) Depth of 30 meters

**Figure 6: Amplitude of velocity for ocean surface waves considering topographic fluctuations.**





Based on the equations (94)-(96), the velocity components in the $x$-axis and $z$-axis $\{u, w\}$, as well as the wave particle trajectory for ocean surface waves, can be described by equation (104) for a single characteristic wave propagating eastward. The horizontal radius appears to be influenced by both topographic fluctuations and current shear, whereas the vertical radius seems to be primarily affected by topographic fluctuations alone.

$$u = \mu_1 \cos(kx - \omega t) \qquad\qquad w = \mu_3 \sin(kx - \omega t)$$

$$\frac{(x - x_0)^2}{\left[\eta\left\{\frac{\cosh\{k(x_3 + H)\}}{\sinh\{kH\}} - \frac{1}{\omega}\frac{\partial U_1}{\partial x_3}\frac{\sinh\{k(x_3 + H)\}}{\sinh\{kH\}}\right\} - \left(\frac{1}{\omega}\right)\mu_{(H_{SM})}k_1(U_1)_{x_3=-H}\left\{\frac{\cosh\{kx_3\}}{\sinh\{kH\}} - \frac{1}{\omega}\frac{\partial U_1}{\partial x_3}\frac{\sinh\{kx_3\}}{\sinh\{kH\}}\right\}\right]^2}$$
$$+ \frac{(z - z_0)^2}{\left[\eta\frac{\sinh\{k(x_3 + H)\}}{\sinh\{kH\}} - \left(\frac{1}{\omega}\right)\left(\mu_{(H_{SM})}k_1(U_1)_{x_3=-H}\right)\frac{\sinh\{kx_3\}}{\sinh\{kH\}}\right]^2} = 1 \qquad (104)$$

The horizontal and vertical radii of wave particle trajectory influenced by topographic fluctuations are shown in Figure 7. Vertical shear of background current $\left\{P = \frac{\partial U_1}{\partial x_3}, \frac{\partial U_2}{\partial x_3}\right\} = 0 s^{-1}$. The horizontal and vertical radii of wave particle trajectory, denoted as $\{Rx, Rz\}$. The variation of the horizontal radius is represented as $\Delta rx = -\left(\frac{1}{\omega}\right)\left(\mu_{(H_{SM})}k_1(U_1)_{x_3=-H}\right)\frac{\cosh(kx_3)}{\sinh(kH)}$, and the variation of the vertical radius is represented as $\Delta rz = -\left(\frac{1}{\omega}\right)\left(\mu_{(H_{SM})}k_1(U_1)_{x_3=-H}\right)\frac{\sinh\{kx_3\}}{\sinh\{kH\}}$. When topographic Fourier coefficient $\mu_{(H_{SM})} = 0.1m$, the variations of radii $\{\Delta rx, \Delta rz\}$ are very small. When topographic Fourier coefficient $\mu_{(H_{SM})} = 0.5m$, even

though there is some modification in horizontal radius at the ocean surface, the impact of topographic fluctuations on the radii of wave particle trajectory primarily occurs at the ocean bottom. In Figure 7 (a3, a4, b3, b4), the radii of wave particle trajectory with a wavelength of 50m exhibit variations up to 0.1-0.2m. Moreover, as background current propagates eastward, horizontal radius diminishes while vertical radius expands, when background current flows westward, horizontal radius enlarges and vertical radius contracts. The phenomenon of gradual enlargement occurs when the radius contracts to zero,

which is a fascinating aspect worthy of investigate. The direction of wave particle velocity may undergo a reversal under this condition.




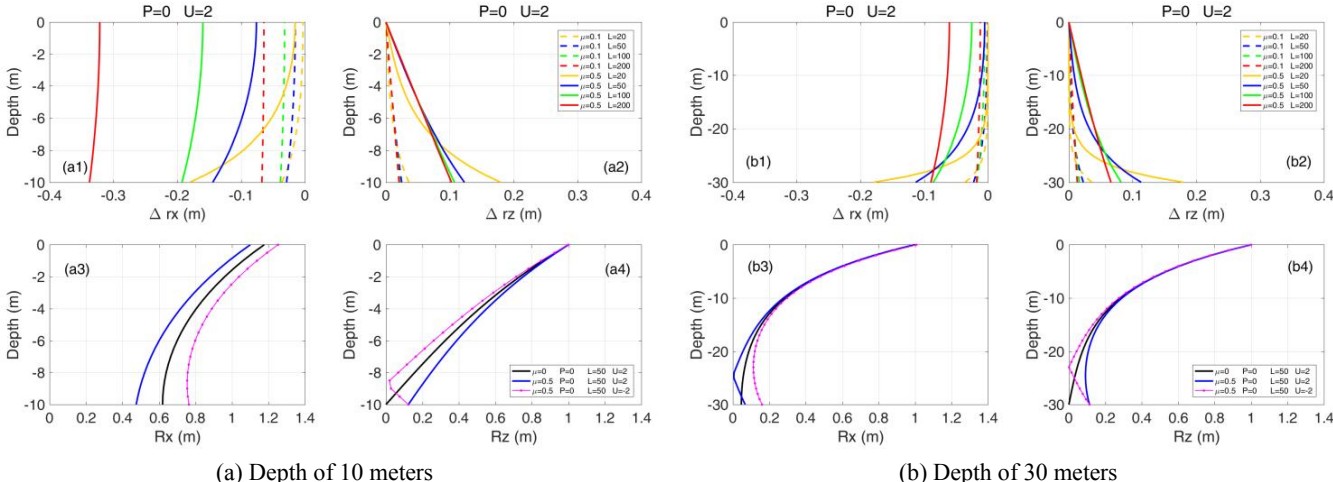

(a) Depth of 10 meters       (b) Depth of 30 meters

**Figure 7: The horizontal and vertical radii for wave particle trajectory considering topographic fluctuations.**

Vertical structure of motion trajectory of one wave particle with a wavelength of 50m is shown in Figure 8, corresponding to Figure 7(a3, a4). The vertical radius of wave particle trajectory, as per the classical solution, remains zero at the ocean
bottom regardless of water depth or background current. The size of the black circles decreases as the depth increases, while the size of the red circles, which account for topographic fluctuations, initially decreases and then increases with increasing depth. The vertical radius of the red circles at the ocean bottom is approximately 0.1m. The modification for vertical radius at a depth of 30m is not significant; therefore, it is omitted in this context.

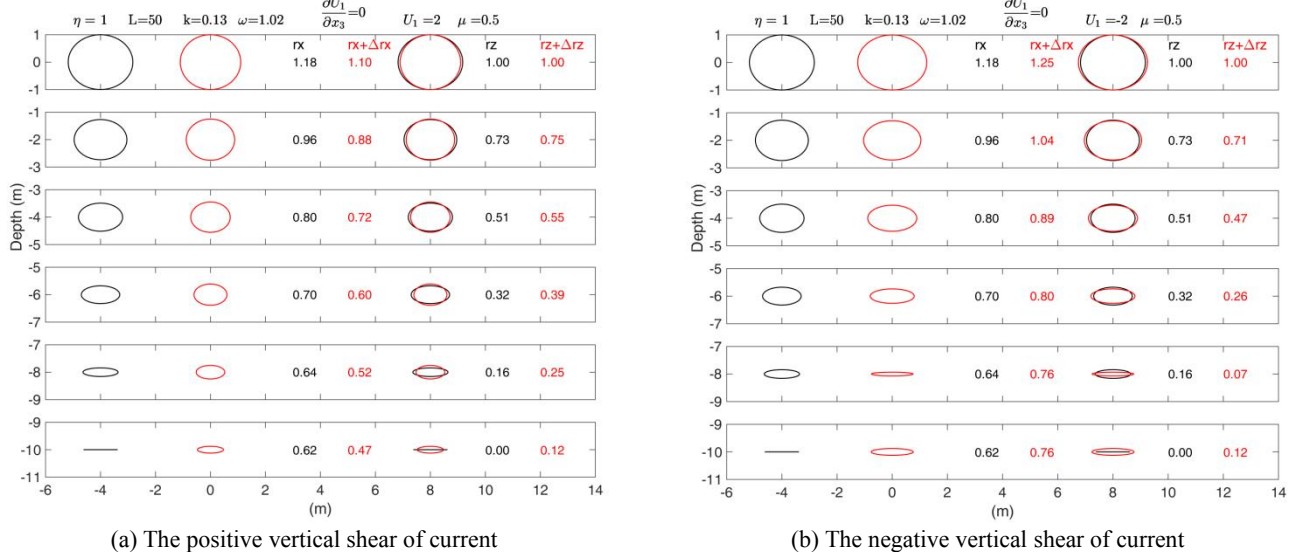

(a) The positive vertical shear of current       (b) The negative vertical shear of current

**Figure 8: Vertical structure of motion trajectory of one wave particle considering topographic fluctuations.**




The horizontal radius of wave particle trajectory, denoted as $Rx$, is influenced by topographic fluctuations and vertical shear of background current. Meanwhile, the vertical radius of wave particle trajectory, denoted as $Rz$, is affected by topographic fluctuations only (refer to Figure 7 and 8). Figure 9 illustrates the influence of topographic fluctuations and vertical shear of background current on the horizontal radius of wave particle trajectory at a depth of 10m. The variation of the horizontal

radius is represented as $\Delta rx = -\left(\frac{1}{\omega}\right)\eta\frac{\partial U_1}{\partial x_3}\frac{\sinh\{k(x_3+H)\}}{\sinh\{kH\}} - \left(\frac{1}{\omega}\right)\mu_{(H_{SM})}k_1(U_1)_{x_3=-H}\left\{\frac{\cosh\{kx_3\}}{\sinh\{kH\}} - \frac{1}{\omega}\frac{\partial U_1}{\partial x_3}\frac{\sinh\{kx_3\}}{\sinh\{kH\}}\right\}$. The comparison

of four cases is illustrated in Figure 9, while the corresponding parameters are presented in Table 1.

**Table 1: Parameters related to topographic fluctuations and background current.**

| Parameters | (a) | (b) | (c) | (d) |
|---|---|---|---|---|
| Topographic Fourier coefficient $\mu_{(H_{SM})}$, (m) | 0 | 0.5 | 0.5 | 0.5 |
| Vertical shear of background current $P = \frac{\partial U_1}{\partial x_3}$, (s$^{-1}$) | 0.2 | 0 | 0.2 | -0.2 |
| Current velocities at the ocean bottom $U_1$, (ms$^{-1}$) | 2 | 2 | 2 | -2 |

According to the comparative analysis of the four cases, it is evident that vertical shear induced by background current plays a pivotal role in reshaping wave particle trajectory at the ocean surface, whereas topographic fluctuations exert a pronounced influence on process occurring at the ocean bottom. The vertical shear of current exerts a greater influence on ocean surface waves characterized by longer wavelengths, whereas topographic fluctuations have a more pronounced impact on ocean surface waves with shorter wavelengths. Several factors, such as the positive and negative values of vertical shear of

background current, current velocities at the ocean bottom, and propagation direction of ocean surface waves,  influence whether these two mechanisms have a weakening or strengthening effect on ocean surface waves.



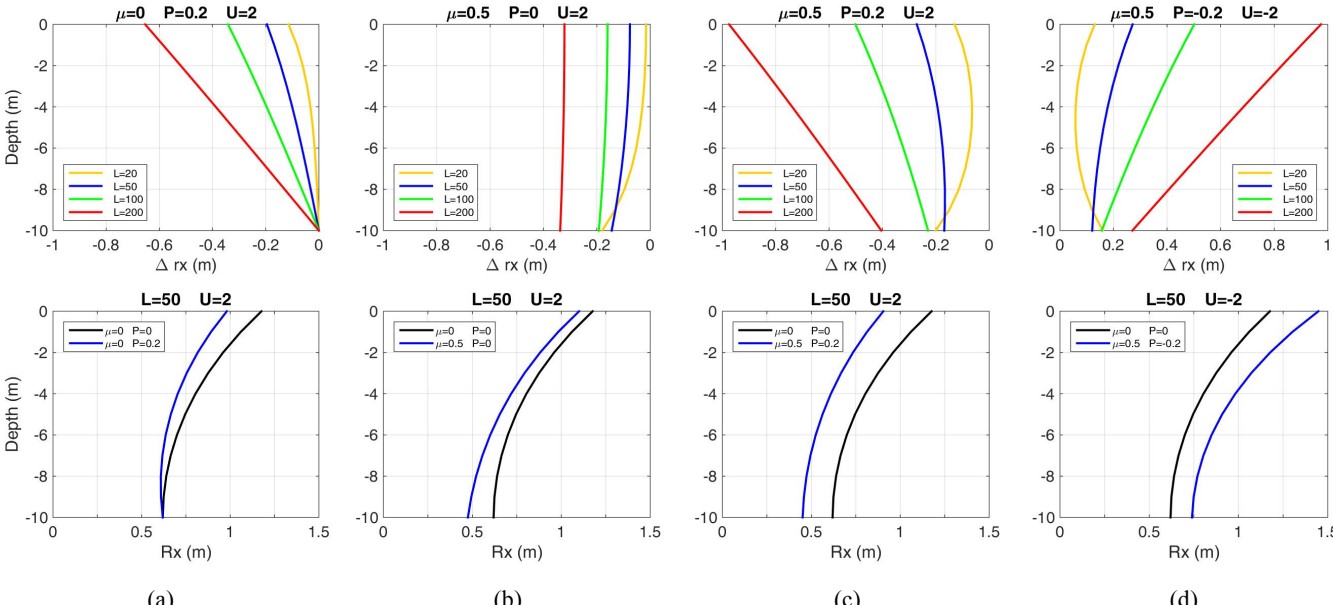

Figure 9: The horizontal radius for wave particle trajectory considering topographic fluctuations and vertical shear of current.

Vertical structure of motion trajectory of one wave particle with a wavelength of 50m is shown in Figure 10, corresponding to Figure 9(c, d). Considering the topographic fluctuations and vertical shear of background current, it is evident that the red

circles exhibits significant dissimilarities compared to the black circles. This disparity may play a important role in the sediment dynamics around coastal regions.

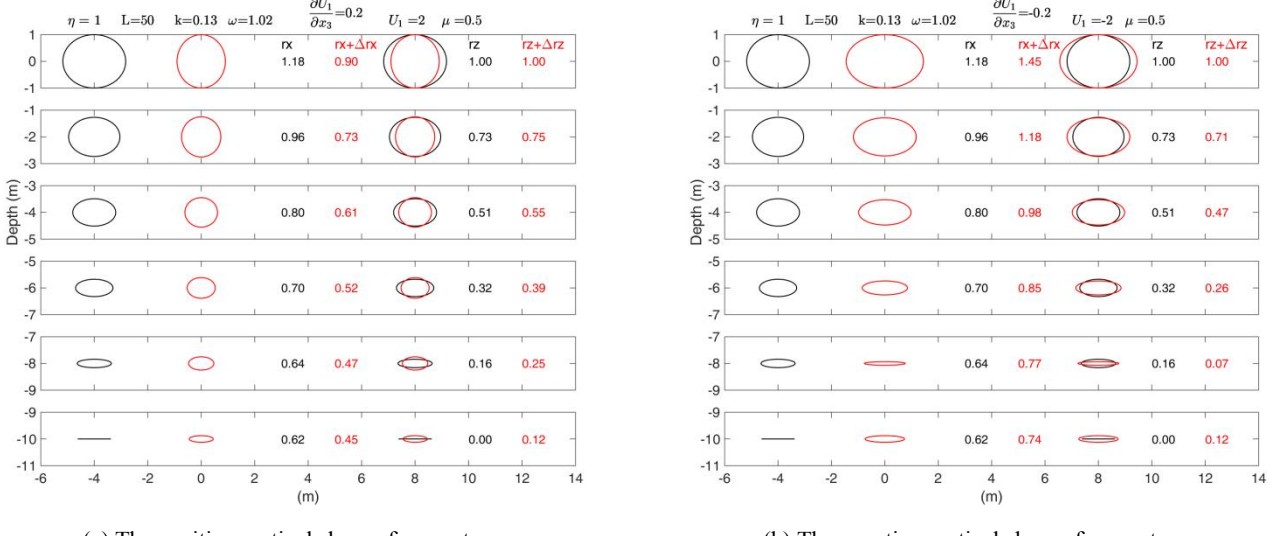

(a) The positive vertical shear of current          (b) The negative vertical shear of current

Figure 10: Vertical structure of motion trajectory of one wave particle considering topographic fluctuations and vertical shear of current.





## 4 Conclusions

The understanding of wave-current interactions is important for comprehending the dynamics of the ocean and accurately forecasting its behaviors. Among them, the mechanisms of vertical shear of background current and topographic fluctuations are the two research issues with limited attention. In this study, we try to evaluate the effect of these two issues on ocean surface waves. Based on the unified wave theory, an analytical model is proposed to describe the modification of the amplitude of orbital velocities for surface waves in presence of background current and topographic fluctuations.


The impacts of vertical shear of background current and topographic fluctuations are quantified on wave particle trajectory in both deep ocean and shallow waters. The principal findings are presented as follows. (1) Vertical shear induced by background current plays a significant role in reshaping wave particle trajectory at the ocean surface, whereas topographic fluctuations exert a pronounced influence on process occurring at the ocean bottom. The modification of the horizontal and

vertical amplitudes of orbital velocities is approximately 0.2-0.3m/s. The horizontal radius of wave particle trajectory in the deep ocean is modified by 0.3m for ocean waves with a wavelength of 200m, while in shallow waters, the vertical radius of wave particle trajectory at the ocean bottom is approximately 0.1m for ocean waves with a wavelength of 50m. (2) The influence of current shear on ocean surface waves is more pronounced for longer wavelengths, whereas topographic fluctuations have a greater impact on ocean surface waves with shorter wavelengths. The interactions between background

current and ocean surface waves are complicated. Under some circumstances, the interaction mechanisms strengthen or weaken the effect on surface waves. In future studies, we will directly validate the proposed analytical model through observations and subsequently incorporate it into a wave numerical model for further application.

## Data availability

Data are available from the authors upon reasonable request.

## Author contributions

YY and MS derived the analytical model. XY designed the experiments and MS carried them out. JD provided multi-beam terrain data. TS and NJ was responsible for visualization. MS prepared the manuscript with contributions from all co-authors.

## Competing interests

The authors declare that they have no conflict of interest.



## Disclaimer

Publisher's note: Copernicus Publications remains neutral with regard to jurisdictional claims made in the text, published maps, institutional affiliations, or any other geographical representation in this paper. While Copernicus Publications makes every effort to include appropriate place names, the final responsibility lies with the authors.

## Acknowledgements

We also thank the anonymous reviewers for useful comments.

## Financial support

This research was jointly funded by the National Key Research Program of China, grant numbers 2022YFC3104800, 2023YFC3008200; Laoshan Laboratory Fund, grant number LSKJ202203003; the National Program on Global Change and Air-Sea Interaction (Phase II), and the Qingdao Postdoctoral Science Foundation, grant number QDBSH2019005.

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
