# Peer review of "An analytical model investigating the impact of current shear and topographic fluctuations on surface waves"

_EGUsphere, 2023_

## Referee Comment (RC2)

**Review report of 'An analytical model investigating the impact of current shear and topographic fluctuations on surface waves'**

by Sun, M. *et al.*

EGUsphere-2023-2905

The manuscript presents an analytical model which, according to the authors, can account for the effects of a vertically sheared current and varying bathymetry on surface water waves. The resulting model is linear and used to examine how a vertically sheared current and slowly varying bathymetry modify the velocities of flow motions induced by linear surface waves in limiting cases.

The paper is written in great detail, which means it is not difficult to follow but the writing is more like a detailed long note than a scientific paper. The referee appreciates the detailed methodology presented by the manuscript which may be of some potential while the final analytical model unfortunately fails to advance much the state-of-the-art. Especially, there have been much simpler approaches that have led to more advanced models than the one presented in the manuscript. Due to the major points given below, the referee would like to recommend the authors to make substantial changes before it warrants publication.

**Major points:**

1. **Introduction**
   There has been an extensive body of recent works being neglected by the authors but relevant to the topic. For instance, Quinn *et al.* (2017) have derived an analytical model using a much simpler approach for surface waves atop a vertically sheared current and slowly varying bathymetry. The model has been validated by Li & Ellingsen (2019). There are also approximate linear models, see, e.g., Ellingsen & Li (2017); Banihashemi & Kirby (2019) among many others, and nonlinear models such as Yang & Liu (2020, 2022); Xin *et al.* (2023); Zheng *et al.* (2023) and references therein.

2. **Analytical derivations presented in §2.1**
   The derivations are made general in that the profile of a background current is permitted to vary in the horizontal plane as well as with depth. The derivations rely on the assumption that the current profile is more slowly varying than the phase of surface waves, although the assumption was not specified.

   (a) The equations from (17) to (68) are sometimes repeated in an unnecessary manner. The key procedures can be summarized into four key step: (1) The coordinate

transformation detailed in Lines 138-140; (2) The Fourier transformation with respect to the horizontal position vector $(x_1, x_2)$) while assuming the current profile to be much slowly varying in the horizontal space and time than the phase of the characteristic waves; (3) elimination of $\mu_1$, $\mu_2$, and $\beta$ in the continuity equation and momentum equation in the vertical direction; and (4) the new definition of $\bar{\mu}_3$ according to (53). With the four key procedures, the derivations can be simplified to a great extent and the writing can be considerably polished. Some of the unnecessary middle steps can be moved to an appendix for curious readers.

(b) The final analytical solution derived is rather disappointing given that complex intermediate steps were used in a reasonable and general manner. It is an analytical model and one would argue that it fails to advance the state of the art due to existing works such as Quinn *et al.* (2017), Li & Ellingsen (2019), and Yang & Liu (2020) and references therein.

(c) The vertical velocity in the form of expression (89) is only valid for a weakly vertically sheared current, see, e.g., Kirby & Chen (1989); Shrira (1993); Ellingsen & Li (2017); Quinn *et al.* (2017). The underlying assumptions associated with the scales of the current profile and waves should thus be made clearer.

(d) How the analytical model advances the state of the art should be mentioned after considering these aforementioned references.

3. **Results**
It would be beneficial to compare the results with these by Quinn *et al.* (2017); Li & Ellingsen (2019) where both a strongly sheared current and a slowly varying bathymetry have been taken into account for describing the linear evolution of surface waves.

**Minor points**

1. Lines 22-27, the description in terms of absolute quantities is very difficult to interpret, e.g., how should one understand a difference by 0.3 m? This difference would be large if it is compared to a magnitude of 1 m but trivial if to a magnitude of 100 m.

2. equation (40) is identical to (49) and (41) is the same as (50). Is it really necessary to list both?

3. 'respectively' was missing in a few places, e.g., the line above eq.(64) and in line 154.

**References**

BANIHASHEMI, S. & KIRBY, J. T. 2019 Approximation of wave action conservation in vertically sheared mean flows. *Ocean Model.* **143**, 101460.

ELLINGSEN, S. Å & LI, Y. 2017 Approximate dispersion relations for waves on arbitrary shear flows. *J. Geophys. Res.: Oceans* **122**, 9889–9905.

KIRBY, J T & CHEN, T 1989 Surface waves on vertically sheared flows: approximate dispersion relations. *J. Geophys. Res. Oceans* **94** (C1), 1013–1027.

LI, Y. & ELLINGSEN, S.Å 2019 A framework for modeling linear surface waves on shear currents in slowly varying waters. *J. Geophys. Res.: Oceans* **124** (4), 2527–2545.

QUINN, BE, TOLEDO, YARON & SHRIRA, VI 2017 Explicit wave action conservation for water waves on vertically sheared flows. *Ocean Model.* **112**, 33–47.

SHRIRA, V. I. 1993 Surface waves on shear currents: solution of the boundary-value problem. *J. Fluid Mech.* **252**, 565–584.

XIN, ZIRUI, LI, XIN & LI, YAN 2023 Coupled effects of wave and depth-dependent current interaction on loads on a bottom-fixed vertical slender cylinder. *Coast. Eng.* p. 104304.

YANG, ZT & LIU, PL-F 2020 Depth-integrated wave–current models. part 1. two-dimensional formulation and applications. *J. Fluid Mech.* **883**, A4.

YANG, ZHENGTONG & LIU, PHILIP L-F 2022 Depth-integrated wave–current models. part 2. current with an arbitrary profile. *J. Fluid Mech.* **936**, A31.

ZHENG, ZIBO, LI, YAN & ELLINGSEN, SIMEN Å 2023 Statistics of weakly nonlinear waves on currents with strong vertical shear. *Phys. Rev. Fluids* **8** (1), 014801.